# Relative Effects of Physical Environment and Employee Performance on Customers' Emotions, Satisfaction, and Behavioral Intentions in Upscale Restaurants

**Kisang Ryu** [1] , **Hyun Jeong Kim** [2] , **Hwangyu Lee** [3],* and **Bongheon Kwon** [4]

1    The College of Hospitality and Tourism Management, Sejong University, Seoul 05006, Korea; kryu11@sejong.ac.kr
2    School of Hospitality Business Management, Carson College of Business, Washington State University, Pullman, WA 99164, USA; jennykim@wsu.edu
3    Law School, Yeungnam University, Gyeongsan 38541, Korea
4    Department of Tourism, Baekseok University, Chungnam 31065, Korea; bongdal@bu.ac.kr
*    Correspondence: slawlhg@naver.com

**Abstract:** This study explored the structural relationships among the physical environment, employee performance, and diners' emotional states, satisfaction, and behavioral intentions, applying the Mehrabian–Russell's theoretical framework in upscale restaurants. Empirical data were collected from 275 upscale restaurant patrons. The results showed that both intangible (employee service) and tangible (physical environment) factors have significant impacts on diners' emotional responses (pleasure and arousal), and these emotional responses affect customer satisfaction and behavioral intentions. This study found that the physical environment exerted a greater impact on arousal than employee behavior while employee behavior had a greater impact on pleasure than physical environment. In addition, arousal was found to have a positive influence on pleasure. We discussed managerial and theoretical implications based on these findings.

**Keywords:** physical environment; employee performance; arousal; pleasure; upscale restaurant

## 1. Introduction

Researchers have actively studied the impact of physical environment on people's emotions using Mehrabian–Russell's [1] theoretical framework, known as the M–R environmental psychology model (hereafter, M–R model). The model suggests that environmental stimuli drive people into certain emotions. In particular, in the hedonic consumption situation, Ryu and Jang [2] reported that physical environment is strongly related to people's emotional response rather than their cognitive evaluations. Thus, the M–R model may be much more prominent to upscale restaurants than casual or quick-service restaurants. While consuming utilitarian services (e.g., Burger King's drive-through) are function-oriented, consuming hedonic services (e.g., dining at upscale restaurants) are emotion-driven [2,3]. Given the possibility of a stronger influence of physical environment on fine dining, more scholarly efforts should be made to examine the role of physical environment in the upscale dining sector.

Most market offerings are a combination of tangible and intangible elements [2], and the restaurant industry is not an exception. Restaurants provide tangibles (e.g., physical environment) and intangibles (e.g., employee performance) to their customers. In addition to the physical environment, the critical role of employee performance has long been recognized in the restaurant business. It is well known employees' service behavior affects customers' emotions/affect, value, customer satisfaction, and their loyalty to restaurants [4,5].

This study was inspired by the following enquiry: "Why are diners willing to pay a higher price in upscale restaurants"? In other words, what are customers seeking through

their fine dining experience? Without any doubt, foods, the most basic element, would play an important role in eliciting customer satisfaction [6,7]. However, we started off this study under the assumption that upscale restaurants generally offer a great food to their patrons. We were rather interested in two other critical factors (human performance and atmosphere) in fine dining where a high degree of hedonic consumption occurs. Although a few studies have investigated the effects of physical environments and employee performance on customer emotions/affects and/or customer satisfaction in restaurants, researchers have focused on one aspect, either the physical environment or human service. In the context of upscale restaurants, both the physical environments and employees' service behavior may significantly affect customers' feelings of pleasure and arousal. To the best of our knowledge, none of the previous studies have answered the following questions: "Out of physical environments and human performance, which one is more critical determinant of pleasure and arousal in the context of upscale restaurants?" In other words, this study attempted to investigate the relative importance of human service performance and atmosphere to customers' two major emotional states (pleasure or arousal) in the upscale restaurant industry.

Additionally, the literature has shown that customer satisfaction is the most important determinant of behavioral intentions [6,7]. However, the original M–R model neglects the key role of employee performance and customer satisfaction because the framework was largely used by retailing scholars whose primary interests lie in how to attract shoppers and extend their length of stay by eliciting positive emotional states. Moreover, the possible causal relationship between the two emotions, themselves (pleasure and arousal) and the possible causal paths between these two emotions and customer satisfaction have not been investigated in the context of upscale restaurants. We, therefore, pursued to address the aforementioned research gaps by modifying the extant M–R model and effectively comparing the influence of employees' service behavior and physical environment, followed by various outcomes, in the setting of luxurious, upscale restaurants.

In summary, the objectives of this study were (1) to examine the construct validity (i.e., convergent validity and discriminant validity) of a modified M–R model after incorporating employee performance; (2) to examine the relative influence of diners' perceptions of the physical environment and employee performance on pleasure and arousal; (3) to test the causal relationship between the two emotional states (pleasure and arousal); (4) to examine the impact of diners' emotions on their satisfaction and behavioral intentions; and (5) to examine the impact of customer satisfaction on behavioral intentions in upscale restaurants.

## 2. Literature Review

### 2.1. Mehrabian–Russell Model

The physical environment refers to the man-made, physical surroundings which are not the natural or social environment [8]. Mehrabian and Russell [1] presented a theoretical framework, explaining the effect of physical environment (also called "servicescape") on human behavior. The core principle of the Mehrabian–Russell (M–R) model is that the physical environment induces one of the three emotions—pleasure/displeasure (e.g., happiness/unhappiness), arousal/non-arousal (e.g., excitement/quiescence), or dominance/submissiveness (e.g., importance/unimportance); people are likely to change their mode of behavior into either approach or avoidance due to the emotional state that they experience in the environment. Simply put, the physical environment has a significant effect on people's emotions and their behavior.

In this study, we defined the physical environment as the man-made physical surroundings in the dining area of upscale restaurants. Restaurant diners want their dining experience to be pleasant, and therefore they look for physical environments that may arouse positive feelings [2]. It is vital for a business to understand customers' emotional responses to a product or service because these emotions may influence customers' purchase decisions. Among three types of emotions (pleasure, arousal, and dominance), there was a non-significant effect of dominance on human behavior; more significant effects

came from pleasure and arousal [9]. In a similar vein, pleasure and arousal were noted as major emotions that lead to positive or negatives responses to the environment in the restaurant setting [2,10]. Pleasure refers to the extent to which individuals feel good, happy, pleased, or joyful in a situation whereas arousal denotes the degree to which individuals feel stimulated, excited, or active [1].

The M–R model has been supported by many empirical studies. For example, Donovan and Rossiter [9] investigated the utility of the M–R model in the formation of positive/negative emotions in the retailing industry. The result showed that store-induced emotion is a powerful predictor of approach-avoidance behaviors and the emotion induced by the store environment affects the extent of extra spending beyond shoppers' original expectations. In the restaurant dining setting, the following two studies are worth attention. Jang and Namkung [10] adopted the M–R model to examine how physical environment influences diners' emotions using full-service restaurant diners. They reported the physical environment as a salient factor that affects customers' emotional responses. Liu and Jang [11] examined the validity of the M–R model and analyzed the empirical data collected from restaurant patrons in the U.S. They found a positive relationship between the physical environment and customer emotion. To sum up, restaurateurs should strive to make a dining ambience attractive to increase diner satisfaction [7,11–13]. Although this M–R model has received a tremendous support in various contexts including, but not limited to, shopping malls, retail outlets, restaurants, and hotels, some prior studies have extended the original M–R model in order to overcome its limitations (e.g., omission of the intangible service aspect, the most crucial determinant of behavioral intentions, i.e., customer satisfaction, and the potential interdependence between pleasure and arousal) [2]. However, none of the previous research has proposed a conceptual model that incorporates all of these limitations in the service industry. Therefore, to fulfil our research goals, we incorporated employee performance and customer satisfaction into the original M–R model in the restaurant industry, particularly the upscale restaurant setting.

## 2.2. Employee Performance

There have been debates about the causal association between employee performance and customer satisfaction. The most common reason for this debate originates from the initial definition of service quality and satisfaction. Perceived service quality (employee performance) was described as a form of a long-term overall evaluation of a product/service while satisfaction was described as a transaction-specific (emotional) evaluation [6,8]. After long debates, the results of empirical studies helped draw a conclusion that service quality is the antecedent of customer satisfaction. Employee performance may generally refer to customers' perception of employees' service behavior during service delivery [14]. This study proposes that diners judge employee performance throughout their meal experience. Therefore, in this study, we defined employee performance as customer perception of employees' service behavior during service delivery at restaurants.

A high level of employee performance has been reported as one of the key factors leading to business success [15–19]. Because human services heavily depend on employees' service skills in the restaurant industry, the interaction between employees and customers exerts a substantial influence on consumers' evaluation towards restaurant services [10]. Reliability, courtesy and knowledge of employees and their willingness to meet customer needs, and their individualized attention to customers can serve as an intangible cue to evaluate employee performance. In restaurant industries, the performance of contact employees is essential to customer perceptions of the restaurant service.

## 2.3. Relationship between Physical Environment and Employee Performance and Emotions

The M–R model facilitates our understanding of the effects of the physical environment on people's emotions and behavior. Prior studies concerning the role of physical environment in customer emotions have revealed the importance of a variety of physical environments—facility aesthetics (e.g., architectural design), layout, and ambience (e.g.,

music, scent, and temperature) and so forth. Wakefield and Blodgett [20] examined customers' response to service quality and atmospherics in three different leisure settings. They discovered that only atmospherics had a positive influence on feelings of excitement (arousal), which in turn led to favorable behavioral intentions (re-patronage intentions and favorable recommendations). Ryu and Jang [2] examined the impact of physical environmental components (facility aesthetics, lighting, ambience, layout, dining equipment, employee appearance) on customers' emotional responses in the upscale restaurant setting. They found that facility aesthetics and employee appearance were significant predictors for pleasure, while employee appearance and ambience were significant predictors for arousal. Ellen and Zhang [21] explored how the restaurant servicescape affected customers' emotional states and behavioral intentions. They found that the restaurant's ambient conditions and signs, symbols, and artifacts had significant effects on the degree of arousal and pleasure experienced by customers.

As for human service, when an employee delivers a friendly service, customers are more likely to feel joy and contentment [22]. For instance, Jang and Namkung [10] found that customer perception of employee performance had a positive effect on positive emotion. Carneiro et al. [23] proposed a conceptual model to examine the indirect effect of eventscape on satisfaction and loyalty via pleasure and arousal (mediators). They found that eventscape (design and entertainment) had significant impacts on both emotions, satisfaction, and loyalty. In accordance with above discussion, we hypothesized that the physical environment and employee performance have positive impacts on pleasure and arousal in the upscale restaurant setting.

**Hypothesis 1a (H1a)**. *The physical environment is positively related to arousal.*

**Hypothesis 1b (H1b)**. *The physical environment is positively related to pleasure.*

**Hypothesis 2a (H2a)**. *Employee performance is positively related to arousal.*

**Hypothesis 2b (H2b)**. *Employee performance is positively related to pleasure.*

### 2.4. Relationships among Customer Emotions, Satisfaction, and Behavioral Intentions

According to the M–R model, the emotional responses to environmental stimuli result in two contrasting behaviors—approach and avoidance [1]. Approach is seen as a positive response that involves a desire for staying, exploring, and affiliating with others in the environment. Avoidance is seen as a negative response that involves a desire to escape from the environment [9]. Given that pleasure and arousal are positive emotions, these emotions can lead people to positive evaluations of the product or service and future buying intentions. In addition, positive emotions are important predictors of customer satisfaction. For example, in Wirtz and Bateson's study [24], pleasure has been found to be strongly related to customer satisfaction. Bigné et al. [25] explored the effects of pleasure and arousal on customer satisfaction in theme parks. They also revealed pleasure as a critical predictor of customer satisfaction. Ryu and Jang [2] found a positive effect of pleasure and arousal on behavioral intensions in upscale restaurants. Jang and Namkung [10] also showed customers' pleasurable feelings as a positive predictor of behavioral intensions in the restaurant industry.

Wakefield and Blodgett [20] reported that the physical environment had a positive impact on the feeling of excitement (arousal), in turn producing re-patronage intentions and favorable recommendations. Ryu and Jang [2] found that although both pleasure and arousal had positive impacts on behavioral intentions, pleasure had a stronger influence on behavioral intentions than arousal. A study by Ellen and Zhang [21] showed that of two emotions (pleasure and arousal), pleasure had a significant influence on behavioral intentions. At the Incheon International Airport, Ryu and Park [26] examined the impact of the experience economy on travelers' pleasure, satisfaction, and airport image; the

researchers revealed that pleasure had a positive impact on satisfaction. Building on the S-O-R (Stimulus–Organism–Response) framework, Nanu et al. [27] discovered that the design of hotel lobby interior had a significant effect on booking intentions across different generations; further, they confirmed a positive influence of emotions on guest satisfaction and behavioral intentions. Carneiro et al. [23] found that the eventscape (design and entertainment) had a significant influence on pleasure and arousal, satisfaction, and loyalty; they further revealed that pleasure was the only emotional dimension that had a significant impact on satisfaction and loyalty and pleasure also served as a mediating variable between the eventscape and satisfaction. In contrast, arousal had neither direct impacts (on satisfaction/loyalty) nor a mediating role (between eventscape and satisfaction). The previous findings do not seem to present certain, definite associations; they rather seem to open up all possible positive relationships between the two emotions and outcomes. Based on the above logic, we developed the hypotheses as follows:

**Hypothesis 3a (H3a)**. *Arousal is positively related to customer satisfaction.*

**Hypothesis 3b (H3b)**. *Arousal is positively related to behavioral intentions.*

**Hypothesis 4a (H4a)**. *Pleasure is positively related to customer satisfaction.*

**Hypothesis 4b (H4b)**. *Pleasure is positively related to behavioral intentions.*

Customer satisfaction has been recognized as a direct determinant of future buying intentions, often used as a surrogate indicator of actual behavior in marketing studies [28–31]. For example, Ryu and Han [6] showed that highly satisfied customers have a strong, positive behavior intentions. Ryu et al. [7] also revealed a positive relationship between customer satisfaction and revisit intentions in restaurant operations. Compared to previous studies, we took a more holistic approach by incorporating all three (emotional states, customer satisfaction, and behavioral intentions) as endogenous outcome variables in the model. We posited the last relational path between customer satisfaction and behavioral intentions in the model as the following:

**Hypothesis 5 (H5)**. *Customer satisfaction is positively related to behavioral intentions.*

### 2.5. Relationship between Arousal and Pleasure

Although the M–R model [1] theorized arousal and pleasure as two orthogonal emotional dimensions. Later studies expressed an oppositional notion against the independency of arousal and pleasure. As evidenced by several studies, a positive relationship between arousal and pleasure has been noted [9,32,33]. For example, Donovan and Rossiter [9] argued that arousal is a key predictor of pleasure. Similarly, Chebat and Michon [33] reported that arousal is a salient factor affecting pleasure. Donovan, Rossiter, and Nesdale [32] concluded that arousal and pleasure are independent yet correlated emotional factors. Based on the above empirical findings, we put forth the following hypothesis:

**Hypothesis 6 (H6)**. *Arousal is positively related to pleasure.*

In summary, this study proposed a conceptual model that hypothesized the relationships among six latent variables in the modified M–R model. The study model is presented in Figure 1.

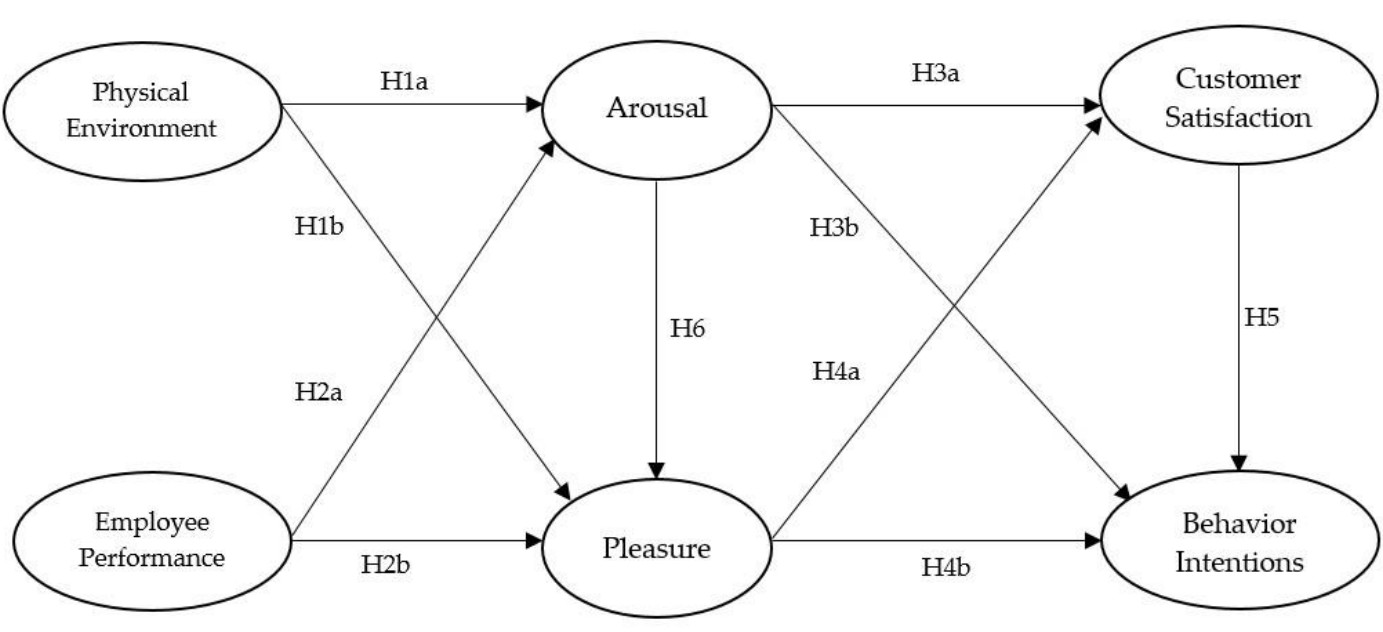

**Figure 1.** The proposed study model.

### 3. Methodology

*3.1. Instruments*

Based on the extensive review of literature [2,34,35], we developed our initial questionnaire including customers' perceived tangible quality (physical environment), intangible quality (employee performance), emotional states, satisfaction, and behavioral intentions. Prior to finalizing the questionnaire, the survey was carefully reviewed by three groups for its content validity: (1) three professors who are familiar with the restaurant industry, (2) graduate students enrolled in the department of hospitality management, and (3) practitioners in the restaurant industry. We conducted a pretest as a preliminary examination of the final version of the survey. Thirty dining customers at an upscale restaurant participated in this pretest to assess the clarity and adequacy of the content. These rigorous steps of content validity resulted in several wording changes.

The items for customer perceptions of physical environment primarily came from the DINESCAPE scale, assessing physical surroundings of the dining room in upscale restaurants [2]. Respondents were asked to rate four items (e.g., "Interior design and décor are visually appealing") on a 7-point Likert scale (1 = strongly disagree and 7 = strongly agree). Employee performance was assessed with five items, focusing on human service behavior (e.g., "Employees were always willing to help me"). A 7-point Likert scale was adopted (1 = strongly disagree and 7 = strongly agree). Emotional states were assessed with seven items [6], representing pleasure (four items) and arousal (three items) recommended by [1]. All items were rated on semantic differentials (scale: −3 ~ +3); these semantic responses were converted into 1 (−3) to 7 (+3) prior to data analysis. Customer satisfaction was assessed on a 7-point Likert scale (1 = strongly disagree and 7 = strongly agree) with three items [6] (e.g., "I have really enjoyed myself at this restaurant"). For behavioral intentions, a scale suggested by Zeithaml et al. [35] was adapted. Participants were asked to respond to three items (e.g., "I would like to come back to this restaurant in the future), using a 7-point Likert scale (1 = strongly disagree and 7 = strongly agree). Finally, participating diners provided their sociodemographic information.

*3.2. Data Collection*

We collected data from the patrons of three upscale restaurants located in Northeast states of the U.S. using a convenience sampling approach. Each restaurant was different

in terms of ownership style (chain or independent) and food offerings. We provided definitions of upscale dining establishments in the survey as follows: restaurants with an average guest check greater than USD 40 and offering exceptional food and service in a luxurious atmosphere. Before data collection, we received permissions from owners or managers. Customers were invited to participate in the study while they were being seated. Those who agreed to participate were given a survey while they were waiting for a check after they finished their main entrée or dessert. Survey administrators explained the purpose of this study to the participating diners. For each restaurant, approximately 150 questionnaires were distributed. In total, 450 questionnaires were distributed, and a total of 300 questionnaires were collected (response rate: 67%). After the elimination of surveys with incomplete responses, 275 questionnaires remained for data analysis [36].

## 4. Data Analysis and Results

### 4.1. Demographic Profile of Respondents

Participants varied in gender (male = 58.1%; female = 41.9%), age ($\leq$25 = 28.8%; 26–35 = 17.6%; 36–45 = 17.3%; 46–55 = 21.3%; $\geq$56 = 15.0%), and household income ($\leq$USD 19,999 = 15.4%; USD 20,000–USD 59,999 = 35.9%; USD 60,000–USD 99,999 = 24.1%; $\geq$USD 100,000 = 24.6%). The majority of participants were Caucasian (87.8%). There were more repeat diners than first time visitors (first time = 45.5%; repeat = 54.5%).

### 4.2. Measurement Model

We tested our proposed model through two steps: a measurement model and a structural model. Confirmatory factor analysis (CFA) was conducted to evaluate how well the measurement model fits the data prior to testing of the structural model. CFA (1) showed the uni-dimensionality of each scale representing the independent concept and (2) validated the measurement model of this study. The fit statistics for the measurement model indicated a good global fit. The ratio ($\chi^2/df$) of 1.89 was below the desired value of 3.0 [37], and the NFI (0.91), IFI (0.96), CFI (0.96), TLI (0.95), and RMSEA (0.05) were within acceptable levels [38]. Table 1 shows study variables with their standardized factor loadings. The factor loadings were equal to or greater than 0.51, and all loadings were significant at $p < 0.001$, with $t$-values ranging from 13.03 to 32.55.

Composite reliability scores (0.87 $\leq$ Reliability $\leq$ 0.96) and average variance extracted (AVE) (0.50 $\leq$ AVEs $\leq$ 0.93) provided evidence for the construct reliability and the convergent validity of all latent variables. Additionally, the squared correlation value between every pair of constructs was lower than the AVE score of each construct in question, indicating the discriminant validity of all latent variables [38]. A summary of the CFA results is shown in Table 2.

### 4.3. Structural Equation Modeling

The proposed model was assessed by the structural equation modeling technique (SEM). The structural model demonstrated satisfactory fit statistics (NFI = 0.91, IFI = 0.95, CFI = 0.95, TLI = 0.94, RMSEA = 0.07) [38]. Figure 2 provides the SEM results with standardized coefficients.

H1a predicted a positive link between physical environment and pleasure. H1a was supported by a significant, positive beta coefficient of 0.46 ($\beta$ = 0.46, $p < 0.05$). H1b, which expected a positive link between physical environment and arousal, was supported by a significant positive beta coefficient of 0.29 ($\beta$ = 0.29, $p < 0.05$). H2a proposed the positive effect of employee performance on pleasure. H2a was supported by a significant, positive beta coefficient of 0.31 ($\beta$ = 0.31, $p < 0.05$). H2b, concerning the positive effect of employee performance on arousal, was supported by a significant, positive beta of 0.36 ($\beta$ = 0.36, $p < 0.05$). In summary, both employee performance and physical environment were found to be influential on upscale restaurant patrons' emotional states (pleasure and arousal).

**Table 1.** Confirmatory factor analysis.

| Construct and Items | Standardized Loading [a] |
|---|---|
| **Physical Environment** | |
| Wall décor is visually appealing. | 0.81 |
| Furniture is of high quality (e.g., chairs and dining tables). | 0.8 |
| Pictures and paintings are attractive. | 0.73 |
| Plaints/flowers make me feel happy. | 0.68 |
| Colors used make me feel warm. | 0.74 |
| Lighting makes me feel welcome. | 0.51 |
| The table setting is visually attractive. | 0.63 |
| **Employee Performance** | |
| Employees serve me at the time promised. | 0.77 |
| Employees quickly correct anything that is wrong. | 0.77 |
| Employees provide prompt and quick service. | 0.77 |
| Employees can answer my questions completely. | 0.74 |
| Employees make me feel comfortable and confident in dealing with them. | 0.79 |
| Employees anticipate my individual needs and wants. | 0.76 |
| Employees seem to have the customers' best interests at heart. | 0.82 |
| **Arousal** | |
| Entertained | 0.77 |
| Excited | 0.88 |
| Surprised | 0.86 |
| **Pleasure** | |
| Happy | 0.81 |
| Pleased | 0.92 |
| Cheerful | 0.89 |
| Delighted | 0.87 |
| **Customer Satisfaction** | |
| I am very satisfied with the overall service at this restaurant. | 0.91 |
| I have really enjoyed myself at this restaurant. | 0.9 |
| **Behavioral Intentions** | |
| I would like to come back to this restaurant in the future. | 0.97 |
| I would recommend this restaurant to my friends or others. | 0.96 |

Note: [a] All loadings are significant at the $p < 0.001$ level.; [b] Bold represents the name of variable.

**Table 2.** Descriptive statistics, correlations, and reliabilities.

| | Mean (SD) | AVE | (1) | (2) | (3) | (4) | (5) | (6) |
|---|---|---|---|---|---|---|---|---|
| (1) Physical Environment | 5.76 (0.53) | 0.5 | 0.87 [a] | 0.51 [b] | 0.58 | 0.59 | 0.65 | 0.59 |
| (2) Employee Performance | 5.98 (0.62) | 0.6 | 0.26 [c] | 0.91 | 0.73 | 0.62 | 0.88 | 0.63 |
| (3) Pleasure | 6.06 (0.63) | 0.76 | 0.34 | 0.53 | 0.93 | 0.74 | 0.84 | 0.7 |
| (4) Arousal | 5.42 (0.96) | 0.71 | 0.34 | 0.38 | 0.55 | 0.88 | 0.69 | 0.54 |
| (5) Customer Satisfaction | 6.16 (0.64) | 0.82 | 0.43 | 0.78 | 0.69 | 0.48 | 0.9 | 0.8 |
| (6) Behavioral Intentions | 6.35 (0.71) | 0.93 | 0.35 | 0.39 | 0.49 | 0.29 | 0.65 | 0.96 |

Notes: SD = Standard Deviation. AVE = Average Variance Extracted. [a] Composite reliabilities are along the diagonal; [b] Correlation coefficients are above the diagonal; [c] Squared correlation coefficients are below the diagonal. All correlation coefficients are significant ($p < 0.05$).

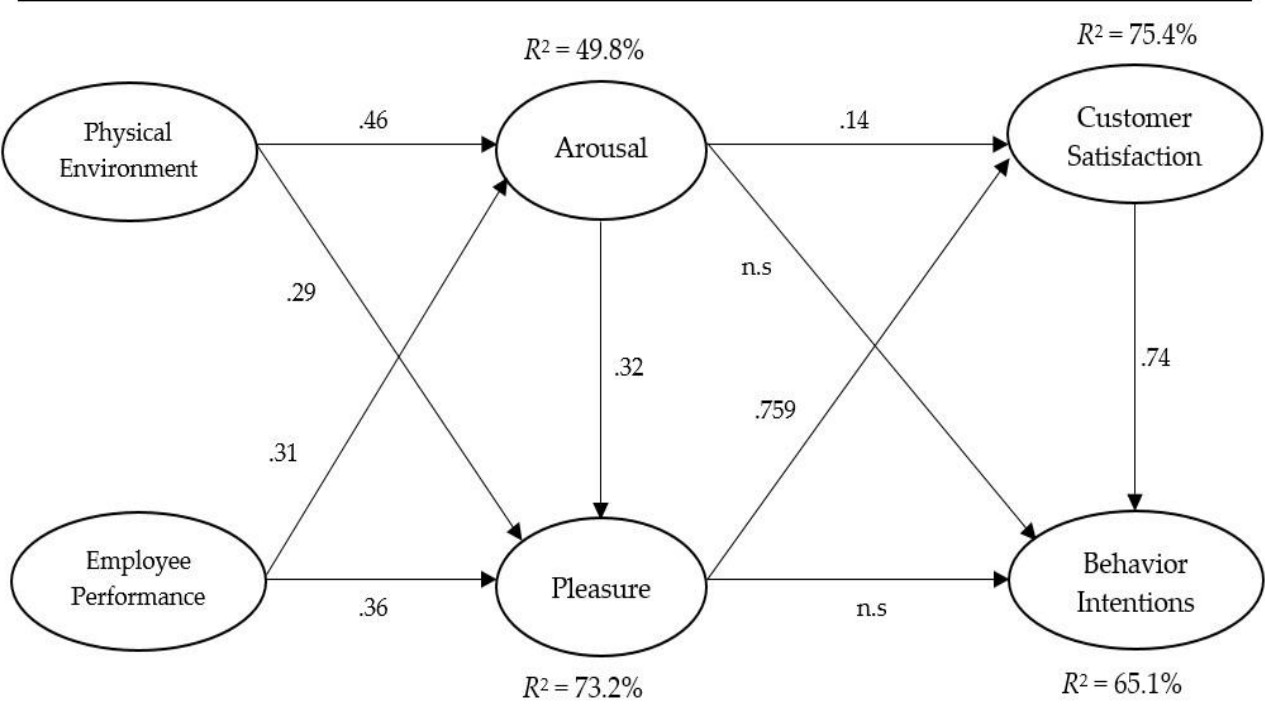

**Figure 2.** SEM results with standardized coefficients.

Then, we proposed the direct influence of diners' emotional states on satisfaction (H3a) and behavioral intentions (H3b). H3a, regarding a positive effect of pleasure on satisfaction, was supported by significant beta coefficient of 0.14 ($\beta = 0.14$, $p < 0.05$); however, there was no significant relationship between arousal and behavioral intentions (H3b). We noted a significant indirect effect of arousal on behavioral intention via satisfaction ($p < 0.05$). H4a regarding a positive relationship between pleasure and customer satisfaction, was supported by a positive beta of 0.76 ($\beta = 0.76$, $p < 0.05$) while H4b, concerning the positive association between pleasure and behavior intentions was not supported. In a similar vein, we noted a significant indirect influence of pleasure on behavior intentions via satisfaction ($p < 0.05$). H5 predicted a positive relationship between customer satisfaction and behavioral intentions. H5 was supported by a significant, positive coefficient of 0.74 ($\beta = 0.74$, $p < 0.05$). Finally, H6, proposing a positive effect of arousal on pleasure, was supported by a significant, positive coefficient of 0.32 ($\beta = 0.74$, $p < 0.05$). In summary, 8 of the 10 hypothesized (direct) paths were significant. The proportion of the variance explained ($R^2$) for the dependent variables were 49.8% for arousal, 73.2% for pleasure, 75.4% for customer satisfaction, and 65.1% for behavioral intentions. These are indicative of large effect sizes.

In addition, Table 3 provides the total effects with the breakdown of direct and indirect effects on outcome variables. First, the direct effects of the physical environment on arousal (0.46) was greater than on pleasure (0.29) while there were no direct effects of the physical environment on customer satisfaction and behavioral intentions. Second, as for indirect effects, the physical environment showed a significant indirect effect on pleasure via arousal (physical environment → arousal → pleasure) ($\beta = 0.15$, $p < 0.05$). Therefore, the total effect (0.44) of the physical environment on pleasure was the combination of significant direct (0.29) and indirect effects (via arousal) (0.15). Third, the direct effect of employee performance on pleasure (0.36) was greater than on arousal (0.31) while there were no direct effects of employee performance on satisfaction and behavioral intentions.

Fourth, regarding indirect effects, employee performance showed a significant indirect effect on pleasure via arousal (employee performance → arousal → pleasure) ($\beta = 0.10$, $p < 0.05$). Therefore, the total effect of employee performance on pleasure (0.46) consisted of significant direct (0.36) and indirect effects (via arousal) (0.10).

**Table 3.** Direct and indirect effects on dependent variables.

| Dependent Variables | Independent Variables | | | | |
|---|---|---|---|---|---|
| | Physical Environment | Employee Performance | Arousal | Pleasure | Customer Satisfaction |
| Arousal | 0.46 [a] (0.46 [b], 0 [c]) | 0.31 (0.31, 0) | - | - | - |
| Pleasure | 0.44 (0.29, 0.15) | 0.46 (0.36, 0.10) | 0.32 (0.32, 0) | - | - |
| Customer satisfaction | 0.40 (0, 0.40) | 0.39 (0, 0.39) | 0.14 (0.14, 0) | 0.76 (0.76, 0) | - |
| Behavioral intentions | 0.29 (0, 0.29) | 0.29 (0, 0.29) | 0.35 (0.07, 0.28) | 0.70 (0.14, 0.56) | 0.74 (0.74, 0) |

Notes. [a]: Total effect; [b]: Direct effect; [c]: Indirect effect.

Next, although no significant direct link (0.07, ns) was found between arousal and behavioral intentions, arousal had a significant, indirect effect on behavior intentions ($\beta = 0.28$, $p < 0.05$) (1) via customer satisfaction (arousal → satisfaction→ behavior intentions) and (2) through longer routes via first pleasure and then customer satisfaction (arousal → pleasure → satisfaction → behavior intentions). Finally, the influence of pleasure on behavioral intentions was also mainly indirect through customer satisfaction (pleasure → satisfaction→ behavioral intentions) ($\beta = 0.56$, $p < 0.05$) with a direct effect being small (0.14). Basically, the majority of total effects of revisit intentions stemming from arousal (0.35) and pleasure (0.70) were accounted for by a mediating variable, customer satisfaction, rather than direct effects.

## 5. Discussions and Implications

Unlike most previous studies, this study attempted to create a modified M–R model by adding an intangible element (employee performance) and customer satisfaction. In the upscale, luxurious dining setting, we purported (1) to investigate the construct validity (i.e., convergent validity and discriminant validity) of a modified M–R model; (2) to compare the relative influence of the physical environment and employee performance on diner emotions (pleasure and arousal); (3) to investigate the interdependence between pleasure and arousal; (4) to examine the impact of emotions on diner satisfaction and behavioral intentions; and (5) to examine the influence of diner satisfaction on behavioral intentions. After proposing these comprehensive theoretical relationships, we used upscale restaurant patrons who were dining at the time of data collection.

The results indicated that the physical environment had a significant impact on diner emotions (arousal and pleasure). These findings are consistent with previous studies [2]. That is, as customers recognize great interior design and décor, lighting, music, and appearance of employees, they are more likely to feel arousal and pleasure. This study confirms the original M–R model, explaining that individuals' emotional states can be affected by physical environments, which in turn elicit a certain behavior.

The salient role of the tangible, physical environment in upscale restaurants should not be neglected. Upscale restaurateurs often invest in professional skills and knowledge of their employees to keep up and improve intangible service quality. Given the emphasis on customer service, upscale restaurant operators may allow the tangible physical environment to deteriorate. This may result in the gradual loss of customers without any recognizable causes. Therefore, operators should monitor customer perceptions of the physical environment (e.g., via a regular online or face-to-face surveys) to find out the need for maintenance, renovation, or remodeling of the establishment. In addition, upscale

restaurateurs must understand what customers are seeking from their dining experience. The physical environment can serve as an effective means to showcase the benefits or value of dining at the upscale restaurant. Operators must identify the key features of their physical environment to enhance the excitement feeling. After that, they must ensure that all the physical details are implemented in a way that is superior to the competition. When other marketing components, such as price or food quality, become neutralized in the fierce competition, especially in the restaurant industry, the physical environment may add a distinctive advantage.

In the modified M–R model, the positive effect of employee service on diner emotions are shown at the upscale restaurant context. We interpret that as diners receive prompt and comfortable services, they are more likely to feel arousal and pleasure. The importance of employee performance has been widely studied in the restaurant industry [5]. They argued that employee service is the key factor for the success of the business because the restaurant industry heavily relies on employees who make constant contacts with customers. Without exception, upscale restaurateurs must maintain the high-quality service at their establishment. Service training is, therefore, a must for employees to learn what excellent customer service looks like. We highly recommend that upscale restaurateurs offer a systematic, regular training rather than a random, irregular training.

One interesting result of this study is that the physical environment exerts a greater impact on arousal than employee performance. This is a crucial finding for upscale restaurant operators. Customers dine out at the upscale restaurant probably to look for the exciting environment, quite different from eating at home. People may feel pleasure at home; however, dining at home may not elicit the feeling of excitement because the environment of their home is no longer interesting. Therefore, customers expect something special from the upscale restaurant. The experience of the upscale restaurant, which is largely driven by the hedonic motive, may make people excited in hopes of escaping their daily routine for a moment, and this could be the reason why people do not mind spending a good amount of money and time at upscale restaurants. Upscale restaurant operators should strive to make their customers entertained, thrilled, and surprised through the development of the physical environment, such as attractive or luxurious decorations, and through a possible seasonal renovation of the physical environment.

In contrast, employee performance played a greater role in pleasure than the physical environment. People generally eat out to satisfy their hunger (utilitarian motive); however, it may be different in the case of the upscale restaurant consumption. Although upscale restaurant patrons, of course, think food is important to quench their hunger, they may focus on happiness and cheerfulness (hedonic motive) while dining. Our result suggests that diners expect employees to help increase a pleasurable dining experience with a friendly, caring service. Therefore, upscale restaurant owners or marketers must advertise the exceptional, personalized service that a diner will receive while dining at their establishment. Electronic word of mouth or online comments (via social media) must be regularly reviewed by the operators to detect any complaints regarding their service.

Another contribution of this study is the clarification of the relative importance of emotions in the segment of upscale restaurants. It is worth noting that pleasure has a greater effect on customer satisfaction than arousal. This sends an important message to upscale restaurateurs. Although diners may be aroused by the physical environment, they ultimately want to feel pleasure. Contact employees should be trained to pay attention to diners' emotional responses to ensure a positive dining experience for each and every customer.

Emotions (arousal and pleasure) were found to be direct predictors for customer satisfaction but not for behavior intentions. Customers' emotional states affected behavior intentions rather indirectly via customer satisfaction. In other words, diners' satisfaction must occur to ensure their revisit. Strictly adopting the M–R theory (emotion to behavior), most prior studies used behavior intentions as the sole outcome of emotions [2,10]. This study extended the prior research by introducing customer satisfaction as a feasible out-

come of emotional states. In fact, the results showed diners' satisfaction as a direct outcome of emotional states, preceding behavioral intentions. Therefore, in the future, hospitality or consumer behavior scholars may want to take customer satisfaction into consideration when explaining the relationship between emotions and behavior intentions.

## 6. Limitations and Future Research

The convenience sampling method limited the generalizability of this study findings. In addition, all study samples came from upscale restaurants in the U.S., representing one particular national culture. To ensure external validity, future research is warranted with diners from other countries to better understand potential cross-cultural behaviors [39]. The results of this study should be validated (tangible and intangible components and their effects on emotions, customer satisfaction, and behavioral intentions) using different samples across various hospitality industries.

Researchers might also want to examine the possible role of demographic differences (e.g., gender and age) because customers' reactions to physical environments and employee performance may differ by demographic characteristics. Finally, prior studies reported the important roles of food quality, price, and location in customers' dining experiences [22]; the relationships between these elements and emotions may provide a promising avenue for future research.

**Author Contributions:** Conceptualization, K.R.; Formal analysis, K.R.; Funding acquisition, H.L.; Methodology, K.R., H.J.K. and B.K.; Supervision, H.L.; Writing—original draft, K.R.; Writing—review & editing, H.J.K. All authors have read and agreed to the published version of the manuscript.

**Funding:** This work was supported by the 2015 Yeungnam University Research Grant.

**Institutional Review Board Statement:** Not applicable.

**Informed Consent Statement:** Not applicable.

**Data Availability Statement:** The data presented in this study are available on request from the corresponding author.

**Conflicts of Interest:** The authors declare no conflict of interest.

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
