# Peer review of "Relative Effects of Physical Environment and Employee Performance on Customers’ Emotions, Satisfaction, and Behavioral Intentions in Upscale Restaurants"

_sustainability, doi:10.3390/su13179549_

Round 1
Reviewer 1 Report
I have enjoyed reviewing this manuscript and in my opinion you could be accepted in its present form.
Author Response
Thank you very much for your generous consideration.
Reviewer 2 Report
Thank you for the opportunity to review the paper “Relative Effects of Physical Environment and Employee Performance on Customers’ Emotional States, Satisfaction, and Behavioral Intentions in Upscale Restaurants”. The topic is interesting and worthy of a wider discussion in the restaurant context.
However, there are many major concerns that need to be addressed, which will be detailed in the following.
Introduction
Lines 37-39. The authors stated that “Given the possibility of a stronger influence of physical environment on fine dining, more scholarly efforts should be made to examine the role of physical environment in the upscale dining sector”. However, given that the literature in this field includes a lot of papers that have addressed similar topics using the M-R model, I suggest the authors explain why it would still be necessary to conduct this research. For example, the authors could highlight the novelty / uniqueness / originality of this research to argue the need / usefulness of this study.
The M-R model as well as constructs (physical environment, employees’ performance, arousal, pleasure, satisfaction and intention) and their relationships are insufficiently documented and described in the Introduction section in order to emphasize the importance of this study. I suggest the authors to improve this section accordingly.
Relatedly, the authors mentioned that “This study modified the M–R theoretical framework by incorporating employee performance in the model in an attempt to test the relationship between employee performance and diners’ emotions (arousal and pleasure).” (lines 343-345). However, I suggest the authors to describe in more detail the original M-R framework as well as the extended framework which they intend to test in the present study (for example, what are the benefits and/or implications of extending the model?)
2.1. Physical environment
In this section the authors aimed to review the literature related to the physical environment, as specified in the title. However, the content of this section is rather related to the M-R model. Accordingly, I suggest the authors either to modify the title in order to reflect the content or modify the content to reflect the title.
In addition, the physical environment dimension needs to be better documented (eventually by adding a new subsection).
The observations and suggestions are identical for section 2.2. Employee performance. For example, the title refers to employee performance, while the content is mostly related to Servqual. In addition, the theory/ model that the authors aimed to investigate in the present study is the M-R model. Consequently, the Servqual technique is irrelevant/ not important for the present research.
2.3. Relationship between physical environment and employee performance and emotions
Further, the authors sought to document the relationship between physical environment and employee performance and emotions. However, emotions were not documented before. Consequently, a new section reviewing the emotions dimension need to be added before this section. In addition, I suggest the authors to move the lines 127-134, 157-163 to this new section since these parts are closely related to emotions.
Hypotheses H1a,b,2a,b, 3a,b, 4a,b were insufficiently documented. Please add more references for each of these hypotheses.
The objectives of this study are not similar in Introduction and Discussion. For example, in Introduction the authors indicate 4 objectives, while in Discussion there are 5 objectives. Please clarify.
The sampling method needs to be described in the Methodology section.
Author Response
Introduction
Comment: Lines 37-39. The authors stated that “Given the possibility of a stronger influence of physical environment on fine dining, more scholarly efforts should be made to examine the role of physical environment in the upscale dining sector”. However, given that the literature in this field includes a lot of papers that have addressed similar topics using the M-R model, I suggest the authors explain why it would still be necessary to conduct this research. For example, the authors could highlight the novelty / uniqueness / originality of this research to argue the need / usefulness of this study.
Response: Thank you for the comments and suggestions to improve the quality of this manuscript. It is true quite a few papers have addressed similar topics using the M-R model in various contexts. However, to the best of our knowledge, none of the previous studies examined the relative impacts of physical environment and employee performance on customer emotions (i.e., pleasure and arousal) in the context of UPSCALE RESTAURANTS. Physical environment plays a key role especially when customers spend moderate-to-long time in a service establishment and when service is consumed primarily for hedonic purposes like dining at upscale restaurant. For instance, at upscale restaurants, customers often spend 2 hours or more, sensing physical surroundings unconsciously and consciously before, during, and after their meal.
None of previous studies have been able to answer the following question: “Out of physical environments and human performance, which one is more significant determinant of pleasure in the context of upscale restaurants?” and “Out of physical environments and human performance, which one is more significant determinant of arousal in the context of upscale restaurants?”
Some prior studies seem to be similar to ours, but they are not the same in many different ways. First, we find a difference in measures of ‘Emotion’. Many prior studies do not follow the measures of ‘Emotion’ suggested by the M-R framework, such as (1) using emotions as a single dimension instead of multiple dimensions, (2) using positive and negative emotions instead of pleasure, arousal (and dominance), (3) using affect or mood instead of emotions, (4) using positive and negative affect instead of emotions. Accordingly, their results can come up with the different interpretation or implications. For instance, Jang and Namkung (2009) found that customer perception of employee performance had a positive effect on positive emotion while it had negative effect on negative emotion. Although they applied M-R model, they used positive emotion and negative emotion instead of pleasure and arousal to represent emotional states like some other studies such as Donovan and Rossiter (1982). Liu and Jang (2009) also showed the validity of the M-R model in the restaurant setting. They analyzed the empirical data collected from restaurant patrons in U.S. and found the positive relationship between physical environment and customer emotion. However, they only used ‘dining atmospherics’ (= physical environment) as a stimulus to influence emotions. Besides this, they used ‘positive emotions’ and ‘negative emotions’ instead of ‘pleasure’ and ‘arousal’ to represent emotional states. They used ‘value’ instead of ‘customer satisfaction’ in their proposed model. Although they are similar, they are definitely not the same in type of sub-dimensions, conceptual definition and operation/measures of variables to capture emotions. Many studies also use different sub-dimensions such as positive affect and negative affect instead of pleasure and arousal. Once again, they are similar. But, they are not the same in type of sub-dimensions, conceptual definition and operation/measures of variables to capture emotions. In sum, this study aimed at providing a comprehensive model by building a conceptual model by applying M-R model (physical environment, arousal, pleasure, behavioral intentions) in conjunction with employee performance and customer satisfaction in restaurant industry, particularly upscale restaurants.
This is the first study to build a conceptual model by applying M-R model (physical environment, arousal, pleasure, behavioral intentions) in conjunction with employee performance and customer satisfaction in restaurant industry, particularly upscale restaurants.
Because of the uniqueness of this study, we came up with some unique findings as follows:
First, this study found that physical environment can be used as more effective marketing tool than employee performance in order to induce arousal such as excitement in upscale restaurants. However, this study found that physical environment and employee performance could make customers feel pleasure positively in similar importance.
Second, pleasure and arousal did not directly affect behavioral intentions unlike the suggestion by M-R model. Instead, pleasure and arousal had positive impact through the mediator (customer satisfaction) in upscale restaurant context. This finding shows the important role of customer satisfaction in the relationship between emotions and behavioral intentions via the customer satisfaction. Therefore, this study suggests that future research should incorporate customer satisfaction onto the original M-R model in the context of upscale restaurants.
Third, this study confirmed the important role of arousal as a mediator between physical environment/employee performance and pleasure and/or customer satisfaction and behavioral intentions through various routs as follows:
Physical environment à arousal → pleasure
Physical environment à arousal → pleasure à customer satisfaction
Physical environment à arousal → pleasure à customer satisfaction à behavior intentions
Physical environment à arousal → customer satisfaction
Physical environment à arousal → customer satisfaction à behavioral intentions
Employee performance à arousal → pleasure
Employee performance à arousal → pleasure à customer satisfaction
Employee performance à arousal → pleasure à customer satisfaction à behavior intentions
Employee performance à arousal → customer satisfaction
Employee performance à arousal → customer satisfaction à behavioral intentions
▶ “In addition, Table III provides total effects with the breakdown of direct and indirect effects on outcome variables. First, the direct effect of physical environment on arousal (.46) was greater than on pleasure (.29) while there were no direct effects of physical environment on customer satisfaction and behavioral intentions. Second, as for indirect effects, physical environment showed a significant indirect effect on pleasure via arousal (physical environment → arousal → pleasure) (β = .15, p < .05). Therefore, the total effect (.44) of physical environment on pleasure was the combination of significant direct (.29) and indirect effects (via arousal) (.15). Third, the direct effect of employee performance on pleasure (.36) was greater than on arousal (.31) while there were no direct effects of employee performance on satisfaction and behavioral intentions. Fourth, regarding indirect effects, employee performance showed a significant indirect effect on pleasure via arousal (employee performance → arousal → pleasure) (β = .10, p < .05). Therefore, the total effect of employee performance on pleasure (.46) consisted of significant direct (.36) and indirect effects (via arousal) (.10).
Next, although no significant direct link (.07, ns) was found between arousal and behavioral intentions, arousal had a significant, indirect effect on behavior intentions (β = .28, p < .05) 1) via customer satisfaction (arousal → satisfaction→ behavior intentions) and 2) through longer routes via first pleasure and then customer satisfaction (arousal → pleasure → satisfaction → behavior intentions). Finally, the influence of pleasure on behavioral intentions was also mainly indirect through customer satisfaction (pleasure → satisfaction→ behavioral intentions) (β = .56, p < .05) with a direct effect being small (.14). Basically, the majority of total effects of revisit intentions stemming from arousal (.35) and pleasure (.70) were accounted for by a mediating variable, customer satisfaction rather than direct effects.”
References
Donovan, R. J.; Rossiter, J. R. Store atmosphere: An environmental psychology approach. Journal of Retailing. 1982, 58, 34-57.
Jang, S.; Namkung, Y. Perceived quality, emotions, and behavioral intentions: Application of an extended Mehrabian-Russell model to restaurants. Journal of Business Research. 2009, 62, 451-460.
Liu, Y.; Jang, S.C.S. The effects of dining atmospherics: An extended Mehrabian-Rusell
model. International Journal of Hospitality Management. 2009, 28, 494-503
▶ “This study found that physical environment exerted a greater impact on arousal than employee behavior while employee behavior had a greater impact on pleasure than physical environment.” (page 1)
▶ “This study was inspired by an enquiry: “Why are diners willing to pay a higher price in the upscale restaurant?” In other words, what are customers seeking through their fine dining experience? Without any doubt, foods, the most basic element would play an important role in eliciting customer satisfaction [6, 7]. However, we started off this study under the assumption that upscale restaurants generally offer a great food to their patrons. We were rather interested in two other critical factors (human performance and atmosphere) in fine dining where a high degree of hedonic consumption occurs. Although a few studies have investigated the effects of physical environments and employee performance on customer emotions/affects and/or customer satisfaction in restaurants, researchers have focused on one aspect, either physical environment or human service. In the context of upscale restaurants, both physical environments and employees’ service behavior may significantly affect customers’ feelings of pleasure and arousal. To the best of our knowledge, none of previous studies have answered the following questions: “Out of physical environments and human performance, which one is more critical determinant of pleasure and arousal in the context of upscale restaurants?” In other words, this study attempted to investigate the relative importance of human service performance and atmosphere to customers’ two major emotional states (pleasure or arousal) in the upscale restaurant industry.
Additionally, the literature has shown that customer satisfaction is the most important determinant of behavioral intentions [6,7]. However, the original M-R model neglects the key role of employee performance and customer satisfaction because the framework was largely used by retailing scholars whose primary interests lie in how to attract shoppers and extend their length of stay by eliciting positive emotional states. Moreover, the possible causal relationship between the two emotions, themselves (pleasure and arousal) and the possible causal paths between these two emotions and customer satisfaction have not been investigated in the context of upscale restaurants. We, therefore, pursued to address the aforesaid research gaps by modifying the extant M-R model and effectively comparing the influence of employees’ service behavior and physical environment, followed by various outcomes, in the setting of luxurious, upscale restaurants.” (page 1-2)
Comment: The M-R model as well as constructs (physical environment, employees’ performance, arousal, pleasure, satisfaction and intention) and their relationships are insufficiently documented and described in the Introduction section in order to emphasize the importance of this study. I suggest the authors to improve this section accordingly.
Response: We revised the Introduction section in order to emphasize the importance of this study in the revised manuscript.
▶ “Most market offerings are a combination of tangible and intangible elements [2], and the restaurant industry is not an exception. Restaurants provide tangibles (e.g., physical environment) and intangibles (e.g., employee performance) to their customers. In addition to physical environment, the critical role of employee performance has long been recognized in the restaurant business. It is well known employees’ service behavior affects customers’ emotions/affect, value, customer satisfaction, and their loyalty to restaurants [4, 5].”
▶ “This study was inspired by an enquiry: “Why are diners willing to pay a higher price in the upscale restaurant?” In other words, what are customers seeking through their fine dining experience? Without any doubt, foods, the most basic element would play an important role in eliciting customer satisfaction [6, 7]. However, we started off this study under the assumption that upscale restaurants generally offer a great food to their patrons. We were rather interested in two other critical factors (human performance and atmosphere) in fine dining where a high degree of hedonic consumption occurs. Although a few studies have investigated the effects of physical environments and employee performance on customer emotions/affects and/or customer satisfaction in restaurants, researchers have focused on one aspect, either physical environment or human service. In the context of upscale restaurants, both physical environments and employees’ service behavior may significantly affect customers’ feelings of pleasure and arousal. To the best of our knowledge, none of previous studies have answered the following questions: “Out of physical environments and human performance, which one is more critical determinant of pleasure and arousal in the context of upscale restaurants?” In other words, this study attempted to investigate the relative importance of human service performance and atmosphere to customers’ two major emotional states (pleasure or arousal) in the upscale restaurant industry.
Additionally, the literature has shown that customer satisfaction is the most important determinant of behavioral intentions [6,7]. However, the original M-R model neglects the key role of employee performance and customer satisfaction because the framework was largely used by retailing scholars whose primary interests lie in how to attract shoppers and extend their length of stay by eliciting positive emotional states. Moreover, the possible causal relationship between the two emotions, themselves (pleasure and arousal) and the possible causal paths between these two emotions and customer satisfaction have not been investigated in the context of upscale restaurants. We, therefore, pursued to address the aforesaid research gaps by modifying the extant M-R model and effectively comparing the influence of employees’ service behavior and physical environment, followed by various outcomes, in the setting of luxurious, upscale restaurants.” (page 1-2)
Comment: Relatedly, the authors mentioned that “This study modified the M–R theoretical framework by incorporating employee performance in the model in an attempt to test the relationship between employee performance and diners’ emotions (arousal and pleasure).” (lines 343-345). However, I suggest the authors to describe in more detail the original M-R framework as well as the extended framework which they intend to test in the present study (for example, what are the benefits and/or implications of extending the model?)
Response: As suggested by the reviewer, we described in more detail the original M-R framework as well as the extended framework which this study intend to test in this study.
▶ “This study was inspired by an enquiry: “Why are diners willing to pay a higher price in the upscale restaurant?” In other words, what are customers seeking through their fine dining experience? Without any doubt, foods, the most basic element would play an important role in eliciting customer satisfaction [6, 7]. However, we started off this study under the assumption that upscale restaurants generally offer a great food to their patrons. We were rather interested in two other critical factors (human performance and atmosphere) in fine dining where a high degree of hedonic consumption occurs. Although a few studies have investigated the effects of physical environments and employee performance on customer emotions/affects and/or customer satisfaction in restaurants, researchers have focused on one aspect, either physical environment or human service. In the context of upscale restaurants, both physical environments and employees’ service behavior may significantly affect customers’ feelings of pleasure and arousal. To the best of our knowledge, none of previous studies have answered the following questions: “Out of physical environments and human performance, which one is more critical determinant of pleasure and arousal in the context of upscale restaurants?” In other words, this study attempted to investigate the relative importance of human service performance and atmosphere to customers’ two major emotional states (pleasure or arousal) in the upscale restaurant industry.
Additionally, the literature has shown that customer satisfaction is the most important determinant of behavioral intentions [6,7]. However, the original M-R model neglects the key role of employee performance and customer satisfaction because the framework was largely used by retailing scholars whose primary interests lie in how to attract shoppers and extend their length of stay by eliciting positive emotional states. Moreover, the possible causal relationship between the two emotions, themselves (pleasure and arousal) and the possible causal paths between these two emotions and customer satisfaction have not been investigated in the context of upscale restaurants. We, therefore, pursued to address the aforesaid research gaps by modifying the extant M-R model and effectively comparing the influence of employees’ service behavior and physical environment, followed by various outcomes, in the setting of luxurious, upscale restaurants.” (page 1-2)
▶ “Physical environment refers to the man-made, physical surroundings, which are not natural or social environment [8]. Mehrabian and Russell [1] presented a theoretical framework, explaining the effect of physical environment (also called “servicescape”) on human behavior. The core principle of Mehrabian-Russell (M-R) model is that physical environment induces one of the three emotions ‒ pleasure/displeasure (e.g., happiness/unhappiness), arousal/non-arousal (e.g., excitement/quiescence), or dominance/submissiveness (e.g., importance/unimportance); and people are likely to change their mode of behavior into either approach or avoidance due to the emotional state that they experience in the environment. Simply put, physical environment has a significant effect on people’s emotions and then their behavior.
In this study, we defined physical environment as man-made physical surroundings in the dining area of upscale restaurants. Restaurant diners want their dining experience to be pleasant, therefore they look for physical environment that may arouse positive feelings [2]. It is vital for a business to understand customers’ emotional responses to a product or service because these emotions may influence customers’ purchase decisions. Among three types of emotions (pleasure, arousal, and dominance), there was a non-significant effect of dominance on human behavior; more significant effects came from pleasure and arousal [9]. In a similar vein, pleasure and arousal were noted as major emotions that lead to positive or negatives responses to the environment in the restaurant setting [2, 10]. Pleasure refers to the extent to which individuals feel good, happy, pleased, or joyful in a situation whereas arousal denotes the degree to which individuals feel stimulated, excited, or active [1].
The M-R model has been supported by many empirical studies. For example, Donovan and Rossiter [9] investigated the utility of the M-R model in the formation of positive/negative emotions in the retailing industry. The result showed that store-induced emotion is a powerful predictor of approach-avoidance behaviors and the emotion induced by the store environment affects the extent of extra spending beyond shoppers’ original expectations. In the restaurant dining setting, the following two studies are worth attention. Jang and Namkung [10] adopted the M-R model to examine how physical environment influences diners’ emotions using full-service restaurant diners. They reported physical environment as a salient factor that affects customers’ emotional responses. Liu and Jang [11] examined the validity of the M-R model and analyzed the empirical data collected from restaurant patrons in the U.S. They found a positive relationship between physical environment and customer emotion. To sum up, restaurateurs should strive to make a dining ambience attractive to increase diner satisfaction [7, 11, 12, 13]. Although this M-R model has received a tremendous support in various contexts including, but not limited to shopping malls, retail outlets, restaurants, and hotels, some prior studies have extended the original M-R model in order to overcome its limitations (e.g., omission of the intangible service aspect, the most crucial determinant of behavioral intentions, i.e., customer satisfaction, and the potential interdependence between pleasure and arousal) [2]. However, none of previous research proposed a conceptual model that incorporated all of these limitation in the service industry. Therefore, to fulfil our research goals, we incorporated employee performance and customer satisfaction into the original M-R model in the restaurant industry, particularly upscale restaurant setting.” (page 2-3)
2.1. Physical environment
Comment: In this section the authors aimed to review the literature related to the physical environment, as specified in the title. However, the content of this section is rather related to the M-R model. Accordingly, I suggest the authors either to modify the title in order to reflect the content or modify the content to reflect the title.
In addition, the physical environment dimension needs to be better documented (eventually by adding a new subsection).
Response: As suggested by the reviewer, we modified the title from ‘Physical environment’ to ‘Mehrabian-Russell model’ and content in order to reflect the content. But, we did not add a new subsection. Rather, we moved the discussion of ‘Emotions’ onto ‘Mehrabian-Russell model.’ We also added more discussion about ‘Emotions.’ Hope the revision can meet your satisfaction.
▶ Mehrabian-Russell (M-R) model [1] provides the theoretical basis for this study to understand the effects of the physical environment on emotional states and the subsequent effect of emotions on behavioral intentions. The fundamental principle of the M-R model is that because physical environment induces people’s emotions such as pleasure/displeasure (e.g., happiness/unhappiness), arousal/non-arousal (e.g., excitement/quiescence), or dominance/submissiveness (e.g., importance/unimportance), people are likely to change their mode of behavior into either approach or avoidance as a result of emotional state that they experience in the environment. Simply put, the physical environment has a significant impact on people’s emotions and then their behavior. Physical environment refers to the man-made, physical surroundings, which are not the natural or social environment [8]. Diners want their dining experience to be pleasant, thus they look for the physical environment that may arouse positive feelings [2]. Mehrabian and Russell [1] presented a theoretical framework, explaining the effect of physical environment (also called “servicescape”) on human behavior. To measure how customers perceive physical environments in upscale restaurants, this study defined it as man-made physical surroundings in the dining area of upscale restaurants.
It is vital for business to understand customers’ emotional responses to a product or service since these emotional responses influence customers’ purchase decisions. As people respond with different sets of emotions to different emotions, the M–R model elicits three components of emotions: pleasure, arousal, and dominance [1]. However, many previous studies found the non-significant effect of dominance on human behavior [9]. In addition, pleasure and arousal dimensions were found to be major responses that lead to positive or negatives responses to the environment in the restaurant setting [2, 10]. Pleasure refers to the extent to which individuals feel good, happy, pleased, or joyful in a situation, whereas arousal denotes the degree to which individuals feel stimulated, excited, or active [1]. In this study, pleasure was defined as the degree to which customers feel happy, pleased, cheerful or delightful in upscale restaurants, while arousal was defied as the extent to which customers entertained, excited, or surprised. The M–R model argues that the emotional responses from environmental stimuli influence two contrasting behaviors (approach and avoidance behavior). That is, emotional responses including pleasure and arousal can result in behavioral intentions.
The M-R model has been supported in many empirical occasions. For example, Donovan and Rossiter [9] investigated the utility of the M-R model in the formation of positive/negative emotions in the retailing industry. The result showed that store-induced emotion is a powerful predictor of approach-avoidance behaviors within the store and the emotion induced by the store environment affects the extent of extra spending beyond shoppers’ original expectations. In the restaurant setting, the following two studies are worthy of noting. Jang and Namkung [10] adopted the M-R model to examine how the physical environment influences diners’ emotions using the data collected from full service restaurant customers. They found physical environment as the salient factor that affects customer emotional responses. Liu and Jang [11] also showed the validity of the M-R model in the restaurant setting. They analyzed the empirical data collected from restaurant patrons in U.S. and found the positive relationship between physical environment and customer emotion. To sum up, prior studies indicated that restaurateurs should strive to make attractive dining ambience to affect diner satisfaction and behavior [7, 11, 12, 13]. Although this M-R model has received consistent support through a lot of empirical studies in various places such as, shopping malls, retail outlets, restaurants, and hotels, many studies have used extended M-R model due to its’ limitation, such as ignoring tangible aspects of service delivery, the most significant determinant (satisfaction) of behavioral intention, and the interdependence between pleasure and arousal [2]. Therefore, this study incorporated ‘employee performance’ and ‘customer satisfaction’ in the original M-R model.”
Comment: The observations and suggestions are identical for section 2.2. Employee performance. For example, the title refers to employee performance, while the content is mostly related to Servqual. In addition, the theory/ model that the authors aimed to investigate in the present study is the M-R model. Consequently, the Servqual technique is irrelevant/ not important for the present research.
Response: Thank you for pointing out the problem in the original manuscript. We largely agree with your concern. Therefore, we largely revised this section in the revised manuscript by deleting the discussion about SERVQUAL and by adding some other rationale.
▶ “There have been mixed findings about the causal direction from employee performance to customer satisfaction. The most common explanation for the difference is that perceived service quality is described as a form of a long-run overall evaluation of a product/service, while satisfaction is described as a transaction-specific evaluation [6, 8]. Employee performance can generally refer to the customer perception of employees’ service behavior during service delivery [14]. This study proposes that customers perceive the employee performance immediately after service experience. Employee performance was defined as customers’ perception of employees’ service behavior during and immediately after service delivery in this study.
A high level of employee performance has been reported as one of the key factors leading to business success [15, 16, 17, 18, 19]. Because human services in the restaurant industry depend heavily on the employees' skills, the interaction between employees and customers can have a substantial influence on the consumers’ evaluation towards restaurant services [10]. Performing the promised service dependably and accurately, courtesy and knowledge of employees and their capability to inspire confidence and trust, employees’ willingness to assist customers and provide prompt service, and employees’ caring and individualized attention can play as intangible cues that create customers’ evaluation about employee performance and customer satisfaction. In restaurant industries, the performance of contact employees is vital to customer perceptions of the restaurant service.”
2.3. Relationship between physical environment and employee performance and emotions
Comment: Further, the authors sought to document the relationship between physical environment and employee performance and emotions. However, emotions were not documented before. Consequently, a new section reviewing the emotions dimension need to be added before this section. In addition, I suggest the authors to move the lines 127-134, 157-163 to this new section since these parts are closely related to emotions.
Response: Thank you for the valuable comment and suggestion. We did not add ‘Emotions’ in the original manuscript since we thought M-R model could discuss ‘Physical environment’ and ‘Emotions.’ Actually, we should have used the title ‘Mehrabian-Russell Model’ instead of ‘Physical environment’ in the original manuscript. However, we did carefully read the content and flow of each section at Literature review after reading your criticism. We realized that we did not do a good job in Literature Review section thanks to your comment. We truly appreciate your valuable comment. Actually, we first decided to add new section (Emotions) in the revised manuscript based on your suggestion. However, on second thought, we decide to combine the writing at ‘Emotion’ section onto the ‘Mehrabian-Russell model.’ We also moved the lines 127-134 to this section since these parts are closely related to emotions as you suggested. However, we did not move the lines 157-163 to this new section since these parts are more closely related to behavioral intentions than emotions.
▶ “Mehrabian-Russell (M-R) model [1] provides the theoretical basis for this study to understand the effects of the physical environment on emotional states and the subsequent effect of emotions on behavioral intentions. The fundamental principle of the M-R model is that because physical environment induces people’s emotions such as pleasure/displeasure (e.g., happiness/unhappiness), arousal/non-arousal (e.g., excitement/quiescence), or dominance/submissiveness (e.g., importance/unimportance), people are likely to change their mode of behavior into either approach or avoidance as a result of emotional state that they experience in the environment. Simply put, the physical environment has a significant impact on people’s emotions and then their behavior. Physical environment refers to the man-made, physical surroundings, which are not the natural or social environment [8]. Diners want their dining experience to be pleasant, thus they look for the physical environment that may arouse positive feelings [2]. Mehrabian and Russell [1] presented a theoretical framework, explaining the effect of physical environment (also called “servicescape”) on human behavior. To measure how customers perceive physical environments in upscale restaurants, this study defined it as man-made physical surroundings in the dining area of upscale restaurants.
It is vital for business to understand customers’ emotional responses to a product or service since these emotional responses influence customers’ purchase decisions. As people respond with different sets of emotions to different emotions, the M–R model elicits three components of emotions: pleasure, arousal, and dominance [1]. However, many previous studies found the non-significant effect of dominance on human behavior [9]. In addition, pleasure and arousal dimensions were found to be major responses that lead to positive or negatives responses to the environment in the restaurant setting [2, 10]. Pleasure refers to the extent to which individuals feel good, happy, pleased, or joyful in a situation, whereas arousal denotes the degree to which individuals feel stimulated, excited, or active [1]. In this study, pleasure was defined as the degree to which customers feel happy, pleased, cheerful or delightful in upscale restaurants, while arousal was defied as the extent to which customers entertained, excited, or surprised. The M–R model argues that the emotional responses from environmental stimuli influence two contrasting behaviors (approach and avoidance behavior). That is, emotional responses including pleasure and arousal can result in behavioral intentions.
The M-R model has been supported in many empirical occasions. For example, Donovan and Rossiter [9] investigated the utility of the M-R model in the formation of positive/negative emotions in the retailing industry. The result showed that store-induced emotion is a powerful predictor of approach-avoidance behaviors within the store and the emotion induced by the store environment affects the extent of extra spending beyond shoppers’ original expectations. In the restaurant setting, the following two studies are worthy of noting. Jang and Namkung [10] adopted the M-R model to examine how the physical environment influences diners’ emotions using the data collected from full service restaurant customers. They found physical environment as the salient factor that affects customer emotional responses. Liu and Jang [11] also showed the validity of the M-R model in the restaurant setting. They analyzed the empirical data collected from restaurant patrons in U.S. and found the positive relationship between physical environment and customer emotion. To sum up, prior studies indicated that restaurateurs should strive to make attractive dining ambience to affect diner satisfaction and behavior [7, 11, 12, 13]. Although this M-R model has received consistent support through a lot of empirical studies in various places such as, shopping malls, retail outlets, restaurants, and hotels, many studies have used extended M-R model due to its’ limitation, such as ignoring tangible aspects of service delivery, the most significant determinant (satisfaction) of behavioral intention, and the interdependence between pleasure and arousal [2]. Therefore, this study incorporated ‘employee performance’ and ‘customer satisfaction’ in the original M-R model.”
Comment: Hypotheses H1a,b,2a,b, 3a,b, 4a,b were insufficiently documented. Please add more references for each of these hypotheses.
Response: We incorporated more up-to-date literature and discussion in order to provide more theoretical and/or logical justification/evidence towards the hypotheses building at Literature Review.
▶ “Physical environment has received a great deal of attention because of its impact on perceptions and emotions. The M–R model facilitates our understanding of the effects of physical environments on emotions and behaviors. Numerous empirical studies about the effect of physical environment on perceptions and emotions have found the impact of physical environment, such as facility aesthetics (e.g., architectural design), layout, and ambience (e.g., music, scent, and temperature) on emotional responses. Wakefield and Blodgett [20] examined customers’ response to service quality (employee performance) and atmospherics (physical environment) in three different leisure settings. The results discovered that atmospherics had a positive influence on feelings of excitement (arousal), which in turn led to favorable behavioral intentions (repatronage intentions and favorable recommendations). Ryu and Jang [2] examined the impact of physical environmental components (facility aesthetics, lighting, ambience, layout, dining equipment, employees) on customers’ emotional responses in the upscale restaurant setting. They found that facility aesthetics and employees were significant predictors for pleasure, while employees and ambience were significant predictors for arousal. Ellen and Zhang [21] explored how restaurant servicescape affected customers’ emotional states and behavioral intentions. This study found that restaurant’s ambient conditions and signs, symbols, and artifacts had significant influence on the degree of arousal and pleasure experienced by the customers. Additionally, pleasure had a significant impact on behavioral intentions.
It has been well recognized that employee performance leads to customers’ emotional responses [10]. In other words, the interaction between customers and service providers is the key factor in the evaluation of overall service quality in the business [22]. For example, when an employee delivers an accurate and friendly service, customers are more likely to feel joy and contentment. Empirical study also showed the effect of employee performance on customers’ emotional responses. For instance, Jang and Namkung [10] have found that customer perception of employee performance had a positive effect on positive emotion while it had negative effect on negative emotion. Carneiro et al. [23] proposed a conceptual model to examine the effect of eventscape of re-enactment events on satisfaction and loyalty by generating emotions (pleasure and arousal) and a mediating effect of both pleasure and arousal between eventscape and satisfaction. This study found that design and entertainment had significant impacts on emotions (both pleasure and arousal), satisfaction, and loyalty. In accordance with this discussion, we hypothesize that physical environment and employee performance have positive impacts on emotions (pleasure and arousal) in the upscale restaurant setting.”
▶ “According to the M–R model, the emotional responses to environmental stimuli result in two contrasting behaviors: approach and avoidance [1]. Approach behaviors are seen as positive responses that involve a desire for staying, exploring, and affiliating with others in the environment. Avoidance behaviors are seen as negative responses that involve a desire for escaping from the environment [9]. In other words, emotional responses including pleasure and arousal can lead to positive evaluations of the product or service and future buying intentions. Some previous studies have found that positive emotions are important predictors of customer satisfaction and behavioral intentions. For example, Wirtz & Bateson [24] showed that pleasure has been found to be strongly related to customer satisfaction. Bigné et al. [25] explored the effects of pleasure and arousal on customer satisfaction in theme parks. They also revealed pleasure as a critical predictor of customer satisfaction. Ryu and Jang [2] found a positive effect of pleasure and arousal on behavior intension in upscale restaurants. Jang and Namkung [10] also showed that customers’ pleasurable feelings positively influence behavior intensions in the restaurant industry. For example, Wakefield and Blodgett [20] found that physical environment had a positive impact on feelings of excitement (arousal), which in turn resulted in repatronage intentions and favorable recommendations. Ryu and Jang [2] found that both pleasure and arousal had positive impacts on behavioral intentions, and pleasure appeared to have a stronger influence than arousal on behavioral intentions. Ellen and Zhang [21] explored how restaurant servicescape affected customers’ emotions (pleasure and arousal) and behavioral intentions, and the results showed that pleasure had a significant influence on behavioral intentions. Ryu and Park [26] examined the impacts of the experience economy of Incheon International Airport on pleasure, satisfaction, and airport image. They revealed that pleasure had a positive impact on satisfaction. Nanu et al. [27] conducted a study to understand what elements of the hotel lobby design affect guest booking intentions in hotels by building on S-O-R (Stimulus-Organism-Response) framework. This study discovered that the lobby interior design style had a significant impact on booking intention across different generations. This study also confirmed the positive effect of emotions on guest satisfaction and behavioral intentions. Carneiro et al. [23] found that design and entertainment had the significant influence on pleasure and arousal, satisfaction, and loyalty. This study also revealed that pleasure was the only dimension of emotions that had a significant impact on satisfaction and loyalty, as well as a mediating effect between eventscape and satisfaction. However, arousal did not have a direct (impact on satisfaction) and mediating role (between eventscape and satisfaction). Based on the aforementioned discussion, we develop the hypotheses as follows:”
Comment: The objectives of this study are not similar in Introduction and Discussion. For example, in Introduction the authors indicate 4 objectives, while in Discussion there are 5 objectives. Please clarify.
Response: As commented by the reviewer, we made the objectives similar in Introduction and Discussion.
▶ “Specifically, the purposes of this study were (1) to examine the construct validity of a modified M–R model after incorporating employee performance, (2) to investigate the relative influence of diners’ perceptions of physical environment and employee performance on pleasure and arousal, (3) to test the causal relationship between pleasure and arousal in the modified M–R model, (4) to examine the impact of diners’ emotions on their satisfaction and behavioral intentions, and (5) to test the impact of customer satisfaction on behavioral intentions in upscale restaurants.” (page 2)
▶ “Unlike previous studies, this study attempted to test a modified M–R model by adding the intangible element (employee performance) and customer satisfaction. This study proposed (1) to investigate the construct validity (i.e., convergent validity and discriminant validity) of a modified M–R model after incorporating employee performance, (2) to investigate the relative influence of customers’ perceptions of physical environment and employee performance on emotions (pleasure and arousal), (3) to test the interdependence between pleasure and arousal in the modified Mehrabian–Russell model, (4) to examine the impact of emotions on customer satisfaction and behavioral intentions; and (5) to test the influence of customer satisfaction on behavioral intentions in the context of upscale restaurants.” (page 9)
Comment: The sampling method needs to be described in the Methodology section.
Response: As suggested by the reviewer, we added more discussion about the sampling method (e.g., convenience sampling approach) in the Methodology section.
▶ “We collected data from the patrons of thee upscale restaurants located in Northeast state of the U.S. using a convenience sampling approach. Each restaurant was different in terms of ownership style (chain or independent) and food offerings. We provided definitions of upscale dining establishments in the survey as follows: restaurants with an average guest check greater than $40, offering exceptional food and service in a luxurious atmosphere. Before data collection, we received permissions from owners or managers. Customers were invited to participate in the study while they were being seated. Those who agreed to participate were given a survey while they were waiting for a check after they finished their main entrée or dessert. Survey administrators explained the purpose of this study to the participating diners. For each restaurant, approximately 150 questionnaires were distributed. In total, 450 questionnaires were distributed, and a total of 300 questionnaires were collected (response rate: 67%). After the elimination of surveys with incomplete responses, 275 questionnaires remained for data analysis [36].” (page 5-6)

Reviewer 3 Report
Relative Effects of Physical Environment and Employee Performance on Customers’ Emotional States, Satisfaction, and Behavioral Intentions in Upscale Restaurants
This is an interesting topic. It can have its merits. However, there are some concerns before I can recommend this manuscript for publication. Here are my comments, concerns and suggestions:
- Please reduce the number of words in your title.
- Avoid citations in your abstract.
- The main question is: “Why are diners willing to pay a (Line46) higher price in the upscale restaurant?” Therefore one would expect that the independent variable would be “Purchase Intention.” However, we do not know that is meant by “Behavior Intentions.”
- The literature review and background section of the study extremely weak. I have no idea about the definition of each variable, and there is not enough evidence presented to show the interrelationship among the variables.
- The hypotheses building sections are very weak. I suggest the authors rewrite the background section.
- We are heavily invested in Mehrabian-Russell Model (MRM). There are more than 1,220 articles since 2017 using this model; however there is little mention about these recent studies. Why did the authors use this model?
- There are different scales for each variable. Some using “7-point Likert scale”and some using “semantic differentials.” Why? Will this be confusing for the respondents? Why not using the same scale?
- The presentation of results with so many tables is redundant. Yet, some crucial information is missing. Please see attached for a more structured table format.
- Table II is confusing. The diagonal should represent square root of AVE, not CR.
- Table II and Table III do not indicate the level of significance, so we do not know if a correlation was significant or not. We do not know if the regression is significant or not.
- Table III should be about mediation analyses. The most important part of this model is about the mediation analysis. Otherwise, the relationship among the variables of the study is very obvious already! We know that physical environment influence arousal, pleasure and satisfaction. Isn’t that very obvious?
- I suggest reconsidering your model. Isn’t true that arousal, and satisfaction are all part of primary emotional response (PER)? See [Ortiz-Ramirez, H. A., Vallejo-Borda, J. A., & Rodriguez-Valencia, A. (2021). Staying on or getting off the sidewalk? Testing the Mehrabian-Russell Model on pedestrian behavior. Transportation research part F: traffic psychology and behaviour, 78, 480-494.]
- Without the mediation effect calculations and proper interpretations, this study doesn’t offer much!
- I suggest rewriting and resubmitting the manuscript.

Author Response
Comment: Please reduce the number of words in your title.
Response: Thank you for your comment. With all due respect, we believe our title perfectly matches the purpose of this study. However, as suggested by the reviewer, we tried to reduce the number of words in the title (albeit not much) from “Relative Effects of Physical Environment and Employee Performance on Customers’ Emotional States, Satisfaction, and Behavioral Intentions in Upscale Restaurants” to “Relative Effects of Physical Environment and Employee Performance on Customers’ Emotions, Satisfaction, and Behavioral Intentions in Upscale Restaurants.” The journal does not have any strict policy regarding the number of words in the title. If you still think we need to reduce the number of words, you can suggest a better title shorter than the current one and feel free to inform us. We, authors would appreciate it.
Comment: Avoid citations in your abstract.
Response: As suggested by the reviewer, we deleted the citation in the abstract.
▶ “This study explored the structural relationships among physical environment, employee performance, diners’ emotional states, satisfaction, and behavioral intentions, applying the Mehrabian-Russell’s theoretical framework in upscale restaurants”
Comment: The main question is: “Why are diners willing to pay a (Line46) higher price in the upscale restaurant?” Therefore one would expect that the independent variable would be “Purchase Intention.” However, we do not know that is meant by “Behavior Intentions.”
Response: We understand why you are asking the question. As you could see in the manuscript, “This study was inspired by an inquiry: “Why are diners willing to pay a higher price in the upscale restaurant?” In other words, what are customers seeking through their fine dining experience?” First of all, “purchase intention” is generally similar to “behavior intention” in conceptual definitions and measures; some researchers interchangeably use them. However, strictly speaking, they are not the same. “Behavior intentions” is broader than “purchase intentions” in the scope. In our study, behavioral intensions were assessed with “revisit intentions” and “recommendation” (see items in Table 1). “Revisit” means “repurchase” at the restaurant setting. We incorporated not only revisit (repurchase) but also one step further, willingness to recommend the restaurant to others. Recommendation is quite typically used by researchers as part of behavioral intention. Behavioral intention is a final dependent variable in our research model ‒ the ultimate goal from restaurant operators’ perspective.
Comment: The literature review and background section of the study extremely weak. I have no idea about the definition of each variable, and there is not enough evidence presented to show the interrelationship among the variables.
Response: We addressed your concerns in the revision. We incorporated more up-to-date literature and discussion in order to make the reader understand better the definition of each variable and the relationships among study variables.
▶ “To measure how customers perceive physical environments in upscale restaurants, this study defined it as man-made physical surroundings in the dining area of upscale restaurants”
▶ “In this study, pleasure was defined as the degree to which customers feel happy, pleased, cheerful or delightful in upscale restaurants, while arousal was defied as the extent to which customers entertained, excited, or surprised”
▶ “Employee performance was defined as customers’ perception of employees’ service behavior during and immediately after service delivery.”
▶ “Physical environment has received a great deal of attention because of its impact on perceptions and emotions. The M–R model facilitates our understanding of the effects of physical environments on emotions and behaviors. Numerous empirical studies about the effect of physical environment on perceptions and emotions have found the impact of physical environment, such as facility aesthetics (e.g., architectural design), layout, and ambience (e.g., music, scent, and temperature) on emotional responses. Wakefield and Blodgett [20] examined customers’ response to service quality (employee performance) and atmospherics (physical environment) in three different leisure settings. The results discovered that atmospherics had a positive influence on feelings of excitement (arousal), which in turn led to favorable behavioral intentions (repatronage intentions and favorable recommendations). Ryu and Jang [2] examined the impact of physical environmental components (facility aesthetics, lighting, ambience, layout, dining equipment, employees) on customers’ emotional responses in the upscale restaurant setting. They found that facility aesthetics and employees were significant predictors for pleasure, while employees and ambience were significant predictors for arousal. Ellen and Zhang [21] explored how restaurant servicescape affected customers’ emotional states and behavioral intentions. This study found that restaurant’s ambient conditions and signs, symbols, and artifacts had significant influence on the degree of arousal and pleasure experienced by the customers. Additionally, pleasure had a significant impact on behavioral intentions.
It has been well recognized that employee performance leads to customers’ emotional responses [10]. In other words, the interaction between customers and service providers is the key factor in the evaluation of overall service quality in the business [22]. For example, when an employee delivers an accurate and friendly service, customers are more likely to feel joy and contentment. Empirical study also showed the effect of employee performance on customers’ emotional responses. For instance, Jang and Namkung [10] have found that customer perception of employee performance had a positive effect on positive emotion while it had negative effect on negative emotion. Carneiro et al. [23] proposed a conceptual model to examine the effect of eventscape of re-enactment events on satisfaction and loyalty by generating emotions (pleasure and arousal) and a mediating effect of both pleasure and arousal between eventscape and satisfaction. This study found that design and entertainment had significant impacts on emotions (both pleasure and arousal), satisfaction, and loyalty. In accordance with this discussion, we hypothesize that physical environment and employee performance have positive impacts on emotions (pleasure and arousal) in the upscale restaurant setting.”
▶ “According to the M–R model, the emotional responses to environmental stimuli result in two contrasting behaviors: approach and avoidance [1]. Approach behaviors are seen as positive responses that involve a desire for staying, exploring, and affiliating with others in the environment. Avoidance behaviors are seen as negative responses that involve a desire for escaping from the environment [9]. In other words, emotional responses including pleasure and arousal can lead to positive evaluations of the product or service and future buying intentions. Some previous studies have found that positive emotions are important predictors of customer satisfaction and behavioral intentions. For example, Wirtz & Bateson [24] showed that pleasure has been found to be strongly related to customer satisfaction. Bigné et al. [25] explored the effects of pleasure and arousal on customer satisfaction in theme parks. They also revealed pleasure as a critical predictor of customer satisfaction. Ryu and Jang [2] found a positive effect of pleasure and arousal on behavior intension in upscale restaurants. Jang and Namkung [10] also showed that customers’ pleasurable feelings positively influence behavior intensions in the restaurant industry. For example, Wakefield and Blodgett [20] found that physical environment had a positive impact on feelings of excitement (arousal), which in turn resulted in repatronage intentions and favorable recommendations. Ryu and Jang [2] found that both pleasure and arousal had positive impacts on behavioral intentions, and pleasure appeared to have a stronger influence than arousal on behavioral intentions. Ellen and Zhang [21] explored how restaurant servicescape affected customers’ emotions (pleasure and arousal) and behavioral intentions, and the results showed that pleasure had a significant influence on behavioral intentions. Ryu and Park [26] examined the impacts of the experience economy of Incheon International Airport on pleasure, satisfaction, and airport image. They revealed that pleasure had a positive impact on satisfaction. Nanu et al. [27] conducted a study to understand what elements of the hotel lobby design affect guest booking intentions in hotels by building on S-O-R (Stimulus-Organism-Response) framework. This study discovered that the lobby interior design style had a significant impact on booking intention across different generations. This study also confirmed the positive effect of emotions on guest satisfaction and behavioral intentions. Carneiro et al. [23] found that design and entertainment had the significant influence on pleasure and arousal, satisfaction, and loyalty. This study also revealed that pleasure was the only dimension of emotions that had a significant impact on satisfaction and loyalty, as well as a mediating effect between eventscape and satisfaction. However, arousal did not have a direct (impact on satisfaction) and mediating role (between eventscape and satisfaction). Based on the aforementioned discussion, we develop the hypotheses as follows:”
Comment: The hypotheses building sections are very weak. I suggest the authors rewrite the background section.”
Response: Hypotheses were discussed in the Literature Review section (we are guessing you meant to say Literature Review section, not the Background section). We revamped the hypotheses sections by providing more theoretical and/or logical justification/evidence (along with more recent literature).
▶ “Physical environment has received a great deal of attention because of its impact on perceptions and emotions. The M–R model facilitates our understanding of the effects of physical environments on emotions and behaviors. Numerous empirical studies about the effect of physical environment on perceptions and emotions have found the impact of physical environment, such as facility aesthetics (e.g., architectural design), layout, and ambience (e.g., music, scent, and temperature) on emotional responses. Wakefield and Blodgett [20] examined customers’ response to service quality (employee performance) and atmospherics (physical environment) in three different leisure settings. The results discovered that atmospherics had a positive influence on feelings of excitement (arousal), which in turn led to favorable behavioral intentions (repatronage intentions and favorable recommendations). Ryu and Jang [2] examined the impact of physical environmental components (facility aesthetics, lighting, ambience, layout, dining equipment, employees) on customers’ emotional responses in the upscale restaurant setting. They found that facility aesthetics and employees were significant predictors for pleasure, while employees and ambience were significant predictors for arousal. Ellen and Zhang [21] explored how restaurant servicescape affected customers’ emotional states and behavioral intentions. This study found that restaurant’s ambient conditions and signs, symbols, and artifacts had significant influence on the degree of arousal and pleasure experienced by the customers. Additionally, pleasure had a significant impact on behavioral intentions.
It has been well recognized that employee performance leads to customers’ emotional responses [10]. In other words, the interaction between customers and service providers is the key factor in the evaluation of overall service quality in the business [22]. For example, when an employee delivers an accurate and friendly service, customers are more likely to feel joy and contentment. Empirical study also showed the effect of employee performance on customers’ emotional responses. For instance, Jang and Namkung [10] have found that customer perception of employee performance had a positive effect on positive emotion while it had negative effect on negative emotion. Carneiro et al. [23] proposed a conceptual model to examine the effect of eventscape of re-enactment events on satisfaction and loyalty by generating emotions (pleasure and arousal) and a mediating effect of both pleasure and arousal between eventscape and satisfaction. This study found that design and entertainment had significant impacts on emotions (both pleasure and arousal), satisfaction, and loyalty. In accordance with this discussion, we hypothesize that physical environment and employee performance have positive impacts on emotions (pleasure and arousal) in the upscale restaurant setting.”
▶ “According to the M–R model, the emotional responses to environmental stimuli result in two contrasting behaviors: approach and avoidance [1]. Approach behaviors are seen as positive responses that involve a desire for staying, exploring, and affiliating with others in the environment. Avoidance behaviors are seen as negative responses that involve a desire for escaping from the environment [9]. In other words, emotional responses including pleasure and arousal can lead to positive evaluations of the product or service and future buying intentions. Some previous studies have found that positive emotions are important predictors of customer satisfaction and behavioral intentions. For example, Wirtz & Bateson [24] showed that pleasure has been found to be strongly related to customer satisfaction. Bigné et al. [25] explored the effects of pleasure and arousal on customer satisfaction in theme parks. They also revealed pleasure as a critical predictor of customer satisfaction. Ryu and Jang [2] found a positive effect of pleasure and arousal on behavior intension in upscale restaurants. Jang and Namkung [10] also showed that customers’ pleasurable feelings positively influence behavior intensions in the restaurant industry. For example, Wakefield and Blodgett [20] found that physical environment had a positive impact on feelings of excitement (arousal), which in turn resulted in repatronage intentions and favorable recommendations. Ryu and Jang [2] found that both pleasure and arousal had positive impacts on behavioral intentions, and pleasure appeared to have a stronger influence than arousal on behavioral intentions. Ellen and Zhang [21] explored how restaurant servicescape affected customers’ emotions (pleasure and arousal) and behavioral intentions, and the results showed that pleasure had a significant influence on behavioral intentions. Ryu and Park [26] examined the impacts of the experience economy of Incheon International Airport on pleasure, satisfaction, and airport image. They revealed that pleasure had a positive impact on satisfaction. Nanu et al. [27] conducted a study to understand what elements of the hotel lobby design affect guest booking intentions in hotels by building on S-O-R (Stimulus-Organism-Response) framework. This study discovered that the lobby interior design style had a significant impact on booking intention across different generations. This study also confirmed the positive effect of emotions on guest satisfaction and behavioral intentions. Carneiro et al. [23] found that design and entertainment had the significant influence on pleasure and arousal, satisfaction, and loyalty. This study also revealed that pleasure was the only dimension of emotions that had a significant impact on satisfaction and loyalty, as well as a mediating effect between eventscape and satisfaction. However, arousal did not have a direct (impact on satisfaction) and mediating role (between eventscape and satisfaction). Based on the aforementioned discussion, we develop the hypotheses as follows:”
Comment: We are heavily invested in Mehrabian-Russell Model (MRM). There are more than 1,220 articles since 2017 using this model; however there is little mention about these recent studies. Why did the authors use this model?
Response: We appreciate the question. Although there is a substantial amount of research about Mehrabian-Russell Model (MRM), this study differs from the previous research in several ways.
▶ “This study found that physical environment exerted a greater impact on arousal than employee behavior while employee behavior had a greater impact on pleasure than physical environment.” (page 1)
▶ “This study was inspired by an enquiry: “Why are diners willing to pay a higher price in the upscale restaurant?” In other words, what are customers seeking through their fine dining experience? Without any doubt, foods, the most basic element would play an important role in eliciting customer satisfaction [6, 7]. However, we started off this study under the assumption that upscale restaurants generally offer a great food to their patrons. We were rather interested in two other critical factors (human performance and atmosphere) in fine dining where a high degree of hedonic consumption occurs. Although a few studies have investigated the effects of physical environments and employee performance on customer emotions/affects and/or customer satisfaction in restaurants, researchers have focused on one aspect, either physical environment or human service. In the context of upscale restaurants, both physical environments and employees’ service behavior may significantly affect customers’ feelings of pleasure and arousal. To the best of our knowledge, none of previous studies have answered the following questions: “Out of physical environments and human performance, which one is more critical determinant of pleasure and arousal in the context of upscale restaurants?” In other words, this study attempted to investigate the relative importance of human service performance and atmosphere to customers’ two major emotional states (pleasure or arousal) in the upscale restaurant industry.
Additionally, the literature has shown that customer satisfaction is the most important determinant of behavioral intentions [6,7]. However, the original M-R model neglects the key role of employee performance and customer satisfaction because the framework was largely used by retailing scholars whose primary interests lie in how to attract shoppers and extend their length of stay by eliciting positive emotional states. Moreover, the possible causal relationship between the two emotions, themselves (pleasure and arousal) and the possible causal paths between these two emotions and customer satisfaction have not been investigated in the context of upscale restaurants. We, therefore, pursued to address the aforesaid research gaps by modifying the extant M-R model and effectively comparing the influence of employees’ service behavior and physical environment, followed by various outcomes, in the setting of luxurious, upscale restaurants.” (page 1-2)
▶ “…… Although this M-R model has received a tremendous support in various contexts including, but not limited to shopping malls, retail outlets, restaurants, and hotels, some prior studies have extended the original M-R model in order to overcome its limitations (e.g., omission of the intangible service aspect, the most crucial determinant of behavioral intentions, i.e., customer satisfaction, and the potential interdependence between pleasure and arousal) [2]. However, none of previous research proposed a conceptual model that incorporated all of these limitation in the service industry. Therefore, to fulfil our research goals, we incorporated employee performance and customer satisfaction into the original M-R model in the restaurant industry, particularly upscale restaurant setting.” (page 2-3)
Comment: There are different scales for each variable. Some using “7-point Likert scale”and some using “semantic differentials.” Why? Will this be confusing for the respondents? Why not using the same scale?
Response: We simply followed original M-R model’s scale. The researchers used the 7-point semantic differential scale to measure emotions (i.e., pleasure, arousal, and dominance) and 7-point Likert scale to measure environmental stimuli and behavior.
Numerous studies have used different scales within one questionnaire. We have not found the theoretical or methodological issue of using different scales in the same questionnaire. In fact, method scholars (e.g., Podsakoff et al., 2003) point out the use of different scales can reduce common method bias (as a way of procedural remedies). In other words, if possible, it is recommended to use a different scale rather than the same scale throughout the questionnaire.
References
Podsakoff, P. M. M., MacKenzie, S. B., Lee, J.-Y., & Podsakoff, N. P. (2003). Common method biases in behavioral research: A critical review of the literature and recommended remedies. Journal of Applied Psychology, 88(5), 879–903.
Comment: The presentation of results with so many tables is redundant. Yet, some crucial information is missing. Please see attached for a more structured table format.
Response: There are only three tables in this study, and all three tables have totally different information (no redundant information). We believe all necessary, essential information is presented in the tables and discussed in the body of the manuscript.
We appreciate you provided a table format to us. However, the table format is appropriate or suitable in this study. Please compare the tables you suggested with our three tables in this study. Your tables are based on both exploratory factor analysis (EFA) and confirmatory factor analysis (CFA). However, we did not conduct EFA; we used SEM, which consists of measurement model (CFA) and structural model (to test our hypotheses). Therefore, we can’t report certain information, such as EFA factor loadings, EFA % of variance, KMO value, or p-value. Instead, we reported information such as standardized factor loadings through CFA.
Besides, all key information is already present in Table 1 and Table 2 in the manuscript. Please take a look at Table 1 and Table 2 carefully. Finally, the information of Table 3, which shows the results of mediation effect (indirect effect) is totally different from the table format you suggested. Therefore, we would like to keep the same tables in the revised manuscript.
Comment: Table II is confusing. The diagonal should represent square root of AVE, not CR.
Response: Our apologies for making you confused. We are aware that diagonal represents AVE in many studies. Diagonal also can represent square root of AVE in few articles. However, there is no standard format to report AVE, CR, or correlations in a table. We have seen various ways the authors report the information. Basically, it is not reasonable to say which way is correct or wrong unless the information of AVE or correlations are missing or calculated wrong. Besides, we reported the information of CR in Table 2. Our notes (under Table 2) clearly state that diagonal represents composite reliabilities (CR) (not AVE). Also, anyone should be able to identify AVE in Table 2 (again, see notes). We believe it is better to report AVE values instead of the square root of AVE because we generally use the AVE value (cut-off point of .5) to see whether or not convergent validity is met or not. One of the main points in Table 2 is whether or not we can identify AVE and correlation coefficients to test construct validity such as convergent and discriminant validities.
▶ Notes: SD = Standard Deviation. AVE = Average Variance Extracted.
a Composite reliabilities are along the diagonal; b Correlation coefficients are above the diagonal; c Squared correlation coefficients are below the diagonal.
All correlation coefficients are significant (p < .05).
Comment: Table II and Table III do not indicate the level of significance, so we do not know if a correlation was significant or not. We do not know if the regression is significant or not.
Response: Based on the reviewer’s comment, we added one more note regarding correlations under Table II (see our previous response as well) and detailed discussion about the significance in the body of the paper to explain Table 3 in the revised manuscript.
▶ “Notes: SD=…………. All correlation coefficients are significant (p < .05).”
▶ “In addition, Table III provides total effects with the breakdown of direct and indirect effects on behavioral intentions. First, the direct effect of physical environment on arousal (.46) is greater than it is on pleasure (.29) while no direct effect of physical environment on customer satisfaction and behavioral intentions exists. Second, in terms of indirect effects, physical environment showed a significant indirect effect on pleasure via arousal (physical environment → arousal → pleasure) (b = .15, p < .05). Therefore, the total effect of physical environment on pleasure (.44) consists of significant direct (.29) and indirect effects (via arousal) (.15). Third, the direct effect of employee performance on pleasure (.36) is greater than it is on pleasure (.31) while no direct effect of employee performance on customer satisfaction and behavioral intentions exists. Fourth, in terms of indirect effects, employee performance showed a significant indirect effect on pleasure via arousal (employee performance → arousal → pleasure) (b = .10, p < .05). Therefore, the total effect of employee performance on pleasure (.46) consists of significant direct (.36) and indirect effects (via arousal) (.10). Next, although no significant direct relationship (.07, ns) was found between arousal and behavioral intentions, arousal had a significant indirect effect on behavior intentions (b = .28, p < .05) via customer satisfaction (arousal → customer satisfaction→ behavior intentions) or through longer routes via first pleasure and then customer satisfaction (arousal → pleasure → customer satisfaction → behavior intentions). Finally, the influence of pleasure on behavioral intentions was also mainly indirect through customer satisfaction (pleasure → arousal → behavioral intentions) (b = .56, p < .05), indicating the majority of total effect of value on revisit intentions (.70) are accounted for by the mediated relationship through satisfaction (.56) rather than the direct effect (.14, ns).”
Comment: Table III should be about mediation analyses. The most important part of this model is about the mediation analysis. Otherwise, the relationship among the variables of the study is very obvious already! We know that physical environment influence arousal, pleasure and satisfaction. Isn’t that very obvious?
Response: Thank you for your comment. However, we would like to express the most important part of this model is not necessarily the mediation analysis. The most critical purpose of this model is to explore the relative impacts of physical environment and employee performance on pleasure and arousal. We also don’t agree with your argument “the relationship among the variables of the study is very obvious already!”.
This study has its own originality/uniqueness accordingly contribution. It is true a number of scholars have addressed similar topics using the M-R model in various contexts. A few studies also examined the role of physical environment in restaurants. However, to the best of our knowledge, none of the previous studies examined the relative impacts of physical environment and employee performance on customer emotions (i.e., pleasure and arousal) in the context of UPSCALE RESTAURANTS. It is important to note that physical environment plays an essential role when customers spend moderate to long time in a service-delivery setting and when service is consumed primarily for hedonic purposes like in upscale restaurants. For instance, customers of upscale restaurants often spend 2 hours or more, sensing physical surroundings unconsciously and consciously before, during, and after their meal.
None of previous studies have been able to answer the following question: “Out of physical environments and human performance, which one is more significant determinant of pleasure in the context of upscale restaurants?” and “Out of physical environments and human performance, which one is more significant determinant of arousal in the context of upscale restaurants?”
None of previous studies have been able to answer the following question: “Out of physical environments and human performance, which one is more significant determinant of pleasure in the context of upscale restaurants?” and “Out of physical environments and human performance, which one is more significant determinant of arousal in the context of upscale restaurants?” Without the findings of this study, none of previous studies can answer these questions. In addition, this is the first study to build a conceptual model by applying M-R model (physical environment, arousal, pleasure, behavioral intentions) in conjunction with employee performance and customer satisfaction in restaurant industry, particularly upscale restaurants. Accordingly, thanks to the originality of this study, this study found a lot of interesting findings.
Some studies in the context of fine dining restaurants seem to be similar. But, they are not the same in some ways. For instance, Jang and Namkung (2009) conducted a study to propose and test a more comprehensive model consisting of three stimuli of perceived quality (product quality, atmospherics, service quality), emotions (positive emotion and negative emotion), and behavioral intentions beyond Mehrabian and Russell's paradigm in full service restaurants (mid-to-upper scale restaurants). They found that customer perception of employee performance had a positive effect on positive emotion while it had negative effect on negative emotion. Although they applied M-R model, they used positive emotion and negative emotion instead of pleasure and arousal to represent emotional states like some other studies such as Donovan and Rossiter (1982). Liu and Jang (2009) also showed the validity of the M-R model in the restaurant setting. They analyzed the empirical data collected from restaurant patrons in U.S. and found the positive relationship between physical environment and customer emotion. However, they only used ‘dining atmospherics’ (= physical environment) as a stimulus to influence emotions. Besides this, they used ‘positive emotions’ and ‘negative emotions’ instead of ‘pleasure’ and ‘arousal’ to represent emotional states. They used ‘value’ instead of ‘customer satisfaction’ in their proposed model. Although they are similar, they are definitely not the same in type of sub-dimensions, conceptual definition and operation/measures of variables to capture emotions. Many studies also use different sub-dimensions such as positive affect and negative affect instead of pleasure and arousal. Once again, they are similar. But, they are not the same in type of sub-dimensions, conceptual definition and operation/measures of variables to capture emotions. In sum, this study aimed at providing a comprehensive model by building a conceptual model by applying M-R model (physical environment, arousal, pleasure, behavioral intentions) in conjunction with employee performance and customer satisfaction in restaurant industry, particularly upscale restaurants.
Moreover, as the reviewer pointed out, the mediation analysis showed a lot interesting results as well. Therefore, as suggested by the reviewer, we added the detailed discussion about the significance within the content to explain Table 3 in the revised manuscript.
▶ “This study found that physical environment exerted a greater impact on arousal than employee behavior while employee behavior had a greater impact on pleasure than physical environment.” (page 1)
▶ “This study was inspired by an enquiry: “Why are diners willing to pay a higher price in the upscale restaurant?” In other words, what are customers seeking through their fine dining experience? Without any doubt, foods, the most basic element would play an important role in eliciting customer satisfaction [6, 7]. However, we started off this study under the assumption that upscale restaurants generally offer a great food to their patrons. We were rather interested in two other critical factors (human performance and atmosphere) in fine dining where a high degree of hedonic consumption occurs. Although a few studies have investigated the effects of physical environments and employee performance on customer emotions/affects and/or customer satisfaction in restaurants, researchers have focused on one aspect, either physical environment or human service. In the context of upscale restaurants, both physical environments and employees’ service behavior may significantly affect customers’ feelings of pleasure and arousal. To the best of our knowledge, none of previous studies have answered the following questions: “Out of physical environments and human performance, which one is more critical determinant of pleasure and arousal in the context of upscale restaurants?” In other words, this study attempted to investigate the relative importance of human service performance and atmosphere to customers’ two major emotional states (pleasure or arousal) in the upscale restaurant industry.
Additionally, the literature has shown that customer satisfaction is the most important determinant of behavioral intentions [6,7]. However, the original M-R model neglects the key role of employee performance and customer satisfaction because the framework was largely used by retailing scholars whose primary interests lie in how to attract shoppers and extend their length of stay by eliciting positive emotional states. Moreover, the possible causal relationship between the two emotions, themselves (pleasure and arousal) and the possible causal paths between these two emotions and customer satisfaction have not been investigated in the context of upscale restaurants. We, therefore, pursued to address the aforesaid research gaps by modifying the extant M-R model and effectively comparing the influence of employees’ service behavior and physical environment, followed by various outcomes, in the setting of luxurious, upscale restaurants.” (page 1-2)
▶ “In addition, Table III provides total effects with the breakdown of direct and indirect effects on outcome variables. First, the direct effect of physical environment on arousal (.46) was greater than on pleasure (.29) while there were no direct effects of physical environment on customer satisfaction and behavioral intentions. Second, as for indirect effects, physical environment showed a significant indirect effect on pleasure via arousal (physical environment → arousal → pleasure) (β = .15, p < .05). Therefore, the total effect (.44) of physical environment on pleasure was the combination of significant direct (.29) and indirect effects (via arousal) (.15). Third, the direct effect of employee performance on pleasure (.36) was greater than on arousal (.31) while there were no direct effects of employee performance on satisfaction and behavioral intentions. Fourth, regarding indirect effects, employee performance showed a significant indirect effect on pleasure via arousal (employee performance → arousal → pleasure) (β = .10, p < .05). Therefore, the total effect of employee performance on pleasure (.46) consisted of significant direct (.36) and indirect effects (via arousal) (.10).
Next, although no significant direct link (.07, ns) was found between arousal and behavioral intentions, arousal had a significant, indirect effect on behavior intentions (β = .28, p < .05) 1) via customer satisfaction (arousal → satisfaction→ behavior intentions) and 2) through longer routes via first pleasure and then customer satisfaction (arousal → pleasure → satisfaction → behavior intentions). Finally, the influence of pleasure on behavioral intentions was also mainly indirect through customer satisfaction (pleasure → satisfaction→ behavioral intentions) (β = .56, p < .05) with a direct effect being small (.14). Basically, the majority of total effects of revisit intentions stemming from arousal (.35) and pleasure (.70) were accounted for by a mediating variable, customer satisfaction rather than direct effects.”
Comment: I suggest reconsidering your model. Isn’t true that arousal, and satisfaction are all part of primary emotional response (PER)? See [Ortiz-Ramirez, H. A., Vallejo-Borda, J. A., & Rodriguez-Valencia, A. (2021). Staying on or getting off the sidewalk? Testing the Mehrabian-Russell Model on pedestrian behavior. Transportation research part F: traffic psychology and behaviour, 78, 480-494.]
Response: Thanks for providing the article. We did read the article. As you already mentioned, there are numerous studies that applied Mehrabian-Russell Model in different contexts. A lot of studies also proposed the extended version of the model by adding variable(s) in the model. Ortiz-Ramirez et al. (2021)’s article is also an example. None of models can be perfect. All models must have limitations. There should be the reason why they apply the original model in the same form or they modify or extend the original model. In Ortiz-Ramirez et al. (2021)’s article, they consider arousal and satisfaction as all part of primary emotional response (PER). Actually, they consider measurement item (satisfaction) as part of “Pleasure” variable, which is one of sub-dimensions of PER. But, this is very rare study that consider satisfaction as part of emotional states. I already acknowledge that some previous studies considered “Satisfaction” as part of Organism in S-O-R (Stimulus-Organism-Response) framework. But, since they did, it does not always mean that we have to follow their rationale. Actually, emotional states are not the same as primary emotional response (PER) in many perspectives such as conceptual definition, measures, sub-dimensions. We are not trying to say that PER is wrong. But, we can definitely argue that PER is not the good way to measure emotional states in the context of upscale restaurants. For instance, ‘pollution’, ‘bike flow’ have nothing to do with “Arousal” in the context of upscale restaurants. I also don’t agree that ‘satisfaction’ to measure “Pleasure” is part of “Pleasure”. Numerous studies show that “Pleasure” and “Satisfaction” are different constructs in definition and measures. For example, we also proved that “Pleasure” and “Satisfaction” are different construct in the conceptual model by using discriminant validity in our study. Similar to most of other studies, we argue that “Satisfaction” is not part of Emotions in our current study. They are different constructs. Therefore, we still keep the same conceptual model in the revision.
References
Ortiz-Ramirez, H. A., Vallejo-Borda, J. A., & Rodriguez-Valencia, A. (2021). Staying on or getting off the sidewalk? Testing the Mehrabian-Russell Model on pedestrian behavior. Transportation Research Part F: Traffic Psychology and Behaviour, 78, 480-494.
Comment: Without the mediation effect calculations and proper interpretations, this study doesn’t offer much!
Response: We understand your point. However, as we responded to you earlier, testing a mediation effect is not the most critical part of this study. Our study has different originality/uniqueness, accordingly contribution.
As you already mentioned, a lot of papers have addressed similar topics using the M-R model in various contexts. Few studies also examined the role of physical environment in restaurants. However, as far as we know, none of the previous studies examined the relative impacts of physical environment and employee performance on customer emotions (i.e., pleasure and arousal) in the context of UPSCALE RESTAURANTS. It is also important to note that physical environment plays an important role when customers spend moderate to long time in a service establishment and when service is consumed primarily for hedonic purposes like dining at upscale restaurants. For instance, at upscale restaurants, customers often spend 2 hours or more, sensing physical surroundings unconsciously and consciously before, during, and after their meal.
None of previous studies have been able to answer the following question: “Out of physical environments and human performance, which one is more critical determinant of pleasure in the context of upscale restaurants?” and “Out of physical environments and human performance, which one is more significant determinant of arousal in the context of upscale restaurants?” Some studies seem to be similar. But, they are not the same in several ways. First, we can find the difference in measures of ‘Emotion”. When scholars apply M-R model, many studies do not follow the measures of ‘Emotion’ suggested by M-R framework, such as (1) using emotions as a single dimension instead of multiple dimensions, (2) using positive and negative emotions, (3) using affect or mood instead of emotions, (4) using positive and negative affect instead of emotions. Accordingly, the results can come up with the different interpretation or implications. For instance, Jang and Namkung (2009) conducted a study to propose and test a more comprehensive model consisting of three stimuli of perceived quality (product quality, atmospherics, service quality), emotions (positive emotion and negative emotion), and behavioral intentions beyond Mehrabian and Russell's paradigm in full service restaurants (mid-to-upper scale restaurants). They found that customer perception of employee performance had a positive effect on positive emotion while it had negative effect on negative emotion. Although they applied M-R model, they used positive emotion and negative emotion instead of pleasure and arousal to represent emotional states like some other studies such as Donovan and Rossiter (1982). Liu and Jang (2009) also showed the validity of the M-R model in the restaurant setting. They analyzed the empirical data collected from restaurant patrons in U.S. and found the positive relationship between physical environment and customer emotion. However, they only used ‘dining atmospherics’ (= physical environment) as a stimulus to influence emotions. Besides this, they used ‘positive emotions’ and ‘negative emotions’ instead of ‘pleasure’ and ‘arousal’ to represent emotional states. They used ‘value’ instead of ‘customer satisfaction’ in their proposed model. Although they are similar, they are definitely not the same in type of sub-dimensions, conceptual definition and operation/measures of variables to capture emotions. Many studies also use different sub-dimensions such as positive affect and negative affect instead of pleasure and arousal. Once again, they are similar. But, they are not the same in type of sub-dimensions, conceptual definition and operation/measures of variables to capture emotions. In sum, this study aimed at providing a comprehensive model by building a conceptual model by applying M-R model (physical environment, arousal, pleasure, behavioral intentions) in conjunction with employee performance and customer satisfaction in restaurant industry, particularly upscale restaurants.
This is the first study to build a conceptual model by applying M-R model (physical environment, arousal, pleasure, behavioral intentions) in conjunction with employee performance and customer satisfaction in restaurant industry, particularly upscale restaurants.
Because of the uniqueness of this study, we came up with some unique findings as follows:
First, this study found that physical environment can be used as more effective marketing tool than employee performance in order to induce arousal such as excitement in upscale restaurants. However, this study found that physical environment and employee performance could make customers feel pleasure positively in similar importance.
Second, pleasure and arousal did not directly affect behavioral intentions unlike the suggestion by M-R model. Instead, pleasure and arousal had positive impact through the mediator (customer satisfaction) in upscale restaurant context. This finding shows the important role of customer satisfaction in the relationship between emotions and behavioral intentions via the customer satisfaction. Therefore, this study suggests that future research should incorporate customer satisfaction onto the original M-R model in the context of upscale restaurants.
Third, this study confirmed the important role of arousal as a mediator between physical environment/employee performance and pleasure and/or customer satisfaction and behavioral intentions through various routs as follows:
Physical environment à arousal → pleasure
Physical environment à arousal → pleasure à customer satisfaction
Physical environment à arousal → pleasure à customer satisfaction à behavior intentions
Physical environment à arousal → customer satisfaction
Physical environment à arousal → customer satisfaction à behavioral intentions
Employee performance à arousal → pleasure
Employee performance à arousal → pleasure à customer satisfaction
Employee performance à arousal → pleasure à customer satisfaction à behavior intentions
Employee performance à arousal → customer satisfaction
Employee performance à arousal → customer satisfaction à behavioral intentions
References
Donovan, R. J.; Rossiter, J. R. Store atmosphere: An environmental psychology approach. Journal of Retailing. 1982, 58, 34-57.
Jang, S.; Namkung, Y. Perceived quality, emotions, and behavioral intentions: Application of an extended Mehrabian-Russell model to restaurants. Journal of Business Research. 2009, 62, 451-460.
Liu, Y.; Jang, S.C.S. The effects of dining atmospherics: An extended Mehrabian-Rusell
model. International Journal of Hospitality Management. 2009, 28, 494-503
Moreover, as the reviewer pointed out, the mediation analysis showed a lot interesting results as well. Therefore, as suggested by the reviewer, we added the detailed discussion about the significance within the content to explain Table 3 in the revised manuscript.
▶ “This study was inspired by an enquiry: “Why are diners willing to pay a higher price in the upscale restaurant?” In other words, what are customers seeking through their fine dining experience? Without any doubt, foods, the most basic element would play an important role in eliciting customer satisfaction [6, 7]. However, we started off this study under the assumption that upscale restaurants generally offer a great food to their patrons. We were rather interested in two other critical factors (human performance and atmosphere) in fine dining where a high degree of hedonic consumption occurs. Although a few studies have investigated the effects of physical environments and employee performance on customer emotions/affects and/or customer satisfaction in restaurants, researchers have focused on one aspect, either physical environment or human service. In the context of upscale restaurants, both physical environments and employees’ service behavior may significantly affect customers’ feelings of pleasure and arousal. To the best of our knowledge, none of previous studies have answered the following questions: “Out of physical environments and human performance, which one is more critical determinant of pleasure and arousal in the context of upscale restaurants?” In other words, this study attempted to investigate the relative importance of human service performance and atmosphere to customers’ two major emotional states (pleasure or arousal) in the upscale restaurant industry.
Additionally, the literature has shown that customer satisfaction is the most important determinant of behavioral intentions [6,7]. However, the original M-R model neglects the key role of employee performance and customer satisfaction because the framework was largely used by retailing scholars whose primary interests lie in how to attract shoppers and extend their length of stay by eliciting positive emotional states. Moreover, the possible causal relationship between the two emotions, themselves (pleasure and arousal) and the possible causal paths between these two emotions and customer satisfaction have not been investigated in the context of upscale restaurants. We, therefore, pursued to address the aforesaid research gaps by modifying the extant M-R model and effectively comparing the influence of employees’ service behavior and physical environment, followed by various outcomes, in the setting of luxurious, upscale restaurants.” (page 1-2)
▶ “In addition, Table III provides total effects with the breakdown of direct and indirect effects on outcome variables. First, the direct effect of physical environment on arousal (.46) was greater than on pleasure (.29) while there were no direct effects of physical environment on customer satisfaction and behavioral intentions. Second, as for indirect effects, physical environment showed a significant indirect effect on pleasure via arousal (physical environment → arousal → pleasure) (β = .15, p < .05). Therefore, the total effect (.44) of physical environment on pleasure was the combination of significant direct (.29) and indirect effects (via arousal) (.15). Third, the direct effect of employee performance on pleasure (.36) was greater than on arousal (.31) while there were no direct effects of employee performance on satisfaction and behavioral intentions. Fourth, regarding indirect effects, employee performance showed a significant indirect effect on pleasure via arousal (employee performance → arousal → pleasure) (β = .10, p < .05). Therefore, the total effect of employee performance on pleasure (.46) consisted of significant direct (.36) and indirect effects (via arousal) (.10).
Next, although no significant direct link (.07, ns) was found between arousal and behavioral intentions, arousal had a significant, indirect effect on behavior intentions (β = .28, p < .05) 1) via customer satisfaction (arousal → satisfaction→ behavior intentions) and 2) through longer routes via first pleasure and then customer satisfaction (arousal → pleasure → satisfaction → behavior intentions). Finally, the influence of pleasure on behavioral intentions was also mainly indirect through customer satisfaction (pleasure → satisfaction→ behavioral intentions) (β = .56, p < .05) with a direct effect being small (.14). Basically, the majority of total effects of revisit intentions stemming from arousal (.35) and pleasure (.70) were accounted for by a mediating variable, customer satisfaction rather than direct effects.”
Comment: I suggest rewriting and resubmitting the manuscript.
Response: Thank you. We resubmit the revised manuscript after addressing all your comments. We appreciate your generous evaluation in advance.
Round 2
Reviewer 2 Report
Now, the paper looks much better. I have not other comments.
Good luck!
Reviewer 3 Report
Relative Effects of Physical Environment and Employee Performance on Customers' Emotions, Satisfaction, and Behavioral Intentions in Upscale Restaurants
Response: Thank you for your comment. With all due respect, we believe our title perfectly matches the purpose of this study.
OK, I have to disagree, but as you wish.
Response: As suggested by the reviewer, we deleted the citation in the abstract.
Thank you.
Response: First of all, "purchase intention" is generally similar to "behavior intention" in conceptual definitions and measures; some researchers interchangeably use them. However, strictly speaking, they are not the same. "Behavior intentions" is broader than "purchase intentions" in the scope.
Where in your paper you clarified this point and what are your documentations/citations for this claim
Response: "Revisit" means "repurchase" at the restaurant setting.
Once again, where did you mention this point? What are your sources? Please refer to the following studies as a sample of recent well-cited articles. None of which uses the term "Behavioral Intentions" the way you use it. Therefore, strong documentation will help the reader believe that you have done your literature review and documented your hard work.
Everett, J. A., Colombatto, C., Chituc, V., Brady, W. J., & Crockett, M. (2020). The effectiveness of moral messages on public health behavioral intentions during the COVID-19 pandemic.
Torlak, N. G., Demir, A., & Budur, T. (2019). Impact of operations management strategies on customer satisfaction and behavioral intentions at café-restaurants. International Journal of Productivity and Performance Management.
Dedeoglu, B. B., Bilgihan, A., Ye, B. H., Buonincontri, P., & Okumus, F. (2018). The impact of servicescape on hedonic value and behavioral intentions: The importance of previous experience. International Journal of Hospitality Management, 72, 10-20.
Response: Recommendation is quite typically used by researchers as part of behavioral intention. Behavioral intention is a final dependent variable in our research model ‒ the ultimate goal from restaurant operators' perspective.
Sorry for the several typos. I meant "dependent variable." Thank you for pointing it out.
Response: We addressed your concerns in the revision.
Thank you. Kindly give the line page number and line number. Definitions should appear right of the term is introduced for the first time. Please also include the source(s) of your definition.
Response: Hypotheses were discussed in the Literature Review section (we are guessing you meant to say Literature Review section, not the Background section).
OK! Thank you for the correction, and thank you for the response.
Response: We appreciate the question. Although there is a substantial amount of research about Mehrabian-Russell Model (MRM), this study differs from the previous research in several ways.
Your response does not address my concern. Your study is different, but other studies have used the same instrument, right? Where are they? Please give reference to the studies that have used the instrument.
Response: We simply followed original M-R model's scale.
OK, thank you! So please give us the information about this Scale and its credentials with references. Podsakoff, P. M. M., MacKenzie, S. B., Lee, J.-Y., & Podsakoff, N. P. (2003) is not enough and is too old as your ONLY reference!
Response: There are only three tables in this study, and all three tables have totally different information (no redundant information). We believe all necessary, essential information is presented in the tables and discussed in the body of the manuscript.
I did not see any information on what software you used.
Response: Our apologies for making you confused. We are aware that diagonal represents AVE in many studies. Diagonal also can represent square root of AVE in few articles. However, there is no standard format to report AVE, CR, or correlations in a table. We have seen various ways the authors report the information.
AVE>MSV is one of the conditions. Your table shows that "2) Employee Performance
does not meet this condition." But you claim: "Additionally, the squared 249 correlation value between every pair of constructs was lower than the AVE"
Response: Based on the reviewer's comment, we added one more note regarding correlations under Table II.
OK! Thank you!
▶ "In addition, Table III provides total effects with the breakdown of direct and indirect effects on behavioral intentions.
Thank you, please also indicate the level of significance. So are we to assume that a,b,c superscripts apply to the whole table?
Response: Thank you for your comment. However, we would like to express the most important part of this model is not necessarily the mediation analysis. The most critical purpose of this model is to explore the relative impacts of physical environment and employee performance on pleasure and arousal. We also don't agree with your argument "the relationship among the variables of the study is very obvious already!"
Thank you! I am afraid I have to disagree with your conclusion. If you are sure about your claim, then please provide studies that contradict the obvious. Please cite some credible studies that indicate that physical Environment and Employee Performance negatively influence Pleasure or Arousal. This should have been part of your literature review. It would be best if you built the case to show that there is a controversy. Thus your study is addressing a key important issue.
" positive effect on positive emotion while it had negative effect on negative emotion." This is a direct relationship, NOT inversed.
The authors have disagreed with the most important issues I brought up. There is not enough citation and documentation to back up the author's several claims, starting with the definition of the terms to model building, choice of instrumentation, and data analyses.